# Reconstruction Alignment Improves Unified Multimodal Models

**Ji Xie[1], Trevor Darrell[1], Luke Zettlemoyer[2], XuDong Wang[1,3]***

[1]UC Berkeley, [2]University of Washington, [3]Duke University

*Project Page:* *https://reconstruction-alignment.github.io/*

## Abstract

Unified multimodal models (UMMs) unify visual understanding and generation within a single architecture. However, conventional training relies on image–text pairs (or sequences) whose captions are typically sparse and miss fine-grained visual details, even when they use hundreds of words to describe a simple image. We introduce **Reconstruction Alignment** (**RECA**), a resource-efficient post-training method that leverages visual understanding encoder embeddings as dense "text prompts," providing rich supervision without captions. Concretely, RECA conditions a UMM on its own visual understanding embeddings and optimizes it to reconstruct the input image with a self-supervised reconstruction loss, thereby realigning understanding and generation. Despite its simplicity, RECA is broadly applicable: across autoregressive, masked-autoregressive, and diffusion-based UMMs, it consistently improves generation and editing fidelity. With only 27 GPU hours, post-training with RECA substantially improves image generation performance on GenEval (0.73→0.90) and DPGBench (80.93→88.15), while also boosting editing benchmarks (ImgEdit 3.38→3.75, GEdit 6.94→7.27). Notably, RECA surpasses much larger open-source models and applies broadly across diverse UMM architectures, establishing it as an efficient and general post-training alignment strategy for UMMs.

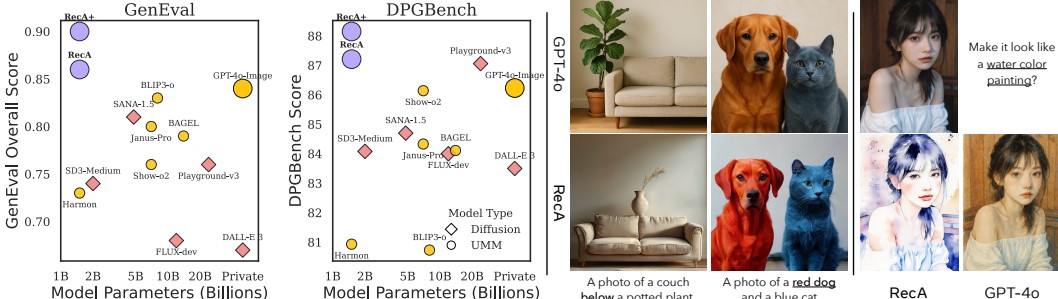

Figure 1: **Post-training UMMs with reconstruction alignment (*i.e.*, RECA) substantially improve image generation and editing.** *Left*: performance comparison on GenEval and DPGBench, where a 1.5B-parameter model post-trained with RECA surpasses much larger models across multiple benchmarks (Table 1: GenEval, DPGBench and WISE); *Middle*: compared with GPT-4o, RECA follows generation instructions more faithfully, especially for *color attributes* and *spatial positions*; *Right*: for editing, RECA better preserves *instance identity*, *overall layout*, and *object shapes* of the original images, such as the girl's lips.

## 1 Introduction

Building on the success of large language models (LLMs) (Brown et al., 2020; Touvron et al., 2023; Yang et al., 2024a), researchers have developed *Multimodal* Large Language Models (MLLMs) (Liu et al., 2024b; Bai et al., 2023; 2025; Radford et al., 2021; Zhai et al., 2023; Chen et al., 2024b; Zhu et al., 2025) with strong visual understanding performance. Recently, unified multimodal models

---

*corresponding author

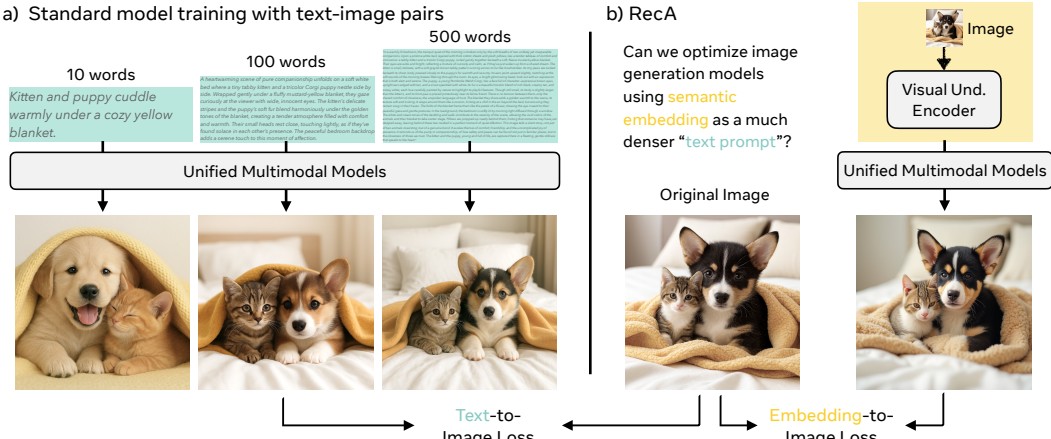

Figure 2: **Dense supervision from visual embeddings.** a) Typical image generation models are trained on image–caption pairs and/or sequences whose text is a *sparse* representation of visual information. ***An image is worth far more than a hundred words*** and contains rich details that text alone cannot capture. As shown in the left three examples, even lengthy captions (500 words) miss key aspects such as *textures, styles, layouts, shapes, and attributes*, leading to imperfect generations relative to the original image. b) By contrast, embeddings from *visual understanding encoders*, *e.g.*, CLIP, preserve richer and more faithful semantics. Can these image–embedding pairs provide the *dense* supervision needed to enhance image generation and editing? Surprisingly, the answer is ***yes***: we find that image–embedding pairs can improve T2I and image editing in a **zero-shot** manner.

(UMMs), or Omni Models, have been proposed to both understand and generate across modalities—reading and writing visual content and text within a single architecture (Tong et al., 2024a; An et al., 2025; Shi et al., 2024; Zhang et al., 2025a; Ge et al., 2024; Wu et al., 2024b; Pan et al., 2025a; OpenAI, 2024; Li et al., 2025; Wu et al., 2025a; Niu et al., 2025a). The academic community envisions that a unified framework can inherit the reasoning and world knowledge of LLMs while extending them to image generation.

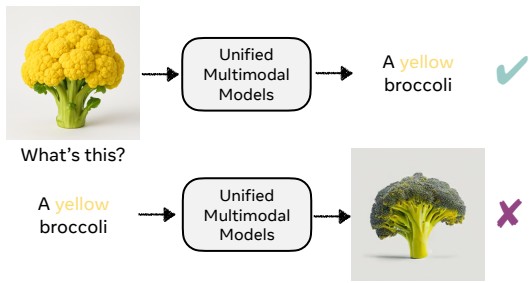

Figure 3: UMMs can often correctly recognize an uncommon concept (yellow broccoli) but fail to generate it, revealing misalignment between understanding and generation.

However, UMMs face a fundamental limitation: conventional training relies on image–text pairs, where text captions provide supervision. Even captions spanning hundreds of words omit critical visual details such as spatial layout, geometry, or fine-grained attributes (Figure 2), introducing systematic biases into the learned representations. For instance, because captions rarely describe broccoli's color, models tend to overfit to the rule *broccoli → green*, often collapsing to green outputs when given atypical prompts like "a *yellow* broccoli" (Figure 3). This misalignment motivates us to explore alternative forms of supervision. Rather than relying on captions, we leverage embeddings from UMM's visual understanding encoders (Radford et al., 2021; Zhai et al., 2023; Chen et al., 2024b; Zhu et al., 2025), which can map pixels into a language-aligned semantic space interpretable by themselves. Crucially, embeddings from *understanding* encoders (*e.g.*, CLIP, SigLIP) capture semantic structure far more effectively than those from *generation* encoders (*e.g.*, VAE, VQ-GAN). These semantic embeddings provide dense, semantically grounded supervision without paired captions, raising a central question: *Can we improve the generation capabilities of UMMs by training them with semantic embeddings as maximally informative "text prompts"?*

Building on these insights, we propose RECA, a resource-efficient post-training strategy. The core idea is simple: condition UMMs on their own visual understanding encoder embeddings—dense "visual prompts" that encode layout, color, and attributes beyond what captions capture—and train them to reconstruct the image. This semantic reconstruction provides richer supervision without

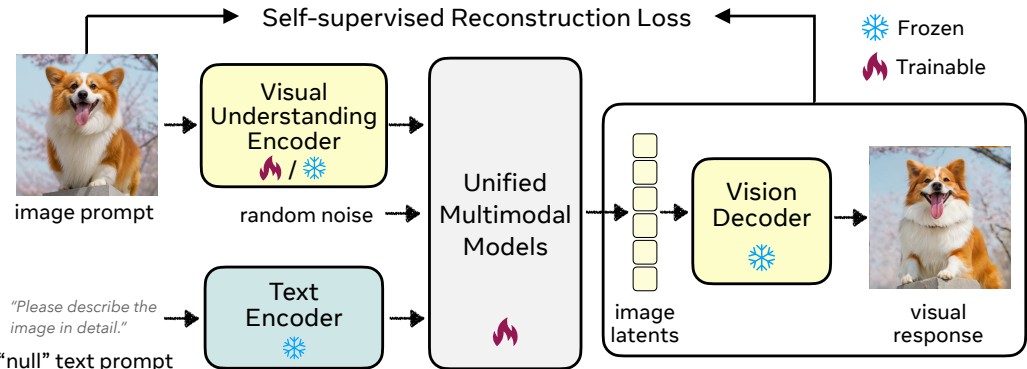

Figure 4: **Overview of the reconstruction alignment (RECA) pipeline**. A visual *understanding* encoder (*e.g.*, CLIP or SigLIP) extracts semantic features from the input image, which are fused with template text embeddings and passed to the Unified Multimodal Model (UMM) to regenerate the image. The UMM is optimized with a self-supervised loss (diffusion or cross-entropy) between the original and generated images or image latents. We **freeze the understanding encoder** except in cases where the UMM employs shared encoder for both understanding and generation (e.g., Harmon). At inference time, RECA **requires no additional inputs**, operating as a standard UMM.

additional labels. Despite its simplicity, RECA yields substantial improvements. With only *27 A100 GPU hours*, a 1.5B-parameter UMM post-trained with RECA surpasses significantly larger open-source models, achieving state-of-the-art results on GenEval (0.86) and DPGBench (87.21). Importantly, these gains are obtained *without any GPT-4o-Image distillation data or reinforcement learning* (OpenAI, 2024; Chen et al., 2025b;c), in contrast to prior work. Moreover, when post-trained with GPT-4o-Image data, RECA further improves to GenEval (0.90) and DPGBench (88.15). RECA also boosts image editing quality, raising ImgEdit from 3.38 to 3.75 and GEdit from 6.94 to 7.27. Moreover, RECA applies broadly across UMM families with different generation mechanisms, including Show-o (Xie et al., 2025b) (Discrete), Harmon (Wu et al., 2025d) (MAR), OpenUni (Wu et al., 2025c) (Continuous), and BAGEL (Deng et al., 2025) (Continuous), highlighting its generality.

The key contributions can be summarized as follows:

- **Method:** We introduce RECA, a semantic reconstruction–based post-training method that uses semantic visual embeddings as "dense prompts", providing rich supervision without captions.
- **Generality:** We show that RECA consistently improves diverse UMM architectures, spanning autoregressive models to hybrid frameworks that integrate autoregressive and diffusion models.
- **Performance:** We demonstrate strong empirical gains, with a 1.5B-parameter model surpassing GPT-4o and larger open-source models using only 27 A100 GPU hours, significantly outperforming prior state-of-the-art methods *without distillation* or *reinforcement learning (RL)*.

## 2 RECONSTRUCTION ALIGNMENT

In this section, we present **Reconstruction Alignment (RECA)** as a self-supervised image reconstruction objective. By training the model to reconstruct images from its visual understanding embeddings, RECA provides dense supervision that captures fine-grained visual details often omitted by text captions. We show the overall pipeline in Figure 4.

### 2.1 MOTIVATION AND SETUP

Equipped with a visual understanding encoder, modern multimodal LLMs (MLLMs) effectively treat images as rich, dense context and successfully handle not only basic perceptual and question answering tasks, but also challenging multimodal reasoning problems such as symbolic, spatial, and mathematical reasoning (Bai et al., 2025; OpenAI, 2024; Huang et al., 2025; Chen et al., 2024a).

One may expect that this dense, text-like context can be naturally extended to the image generation task. However, when we extract embeddings from the understanding encoder, insert them into a

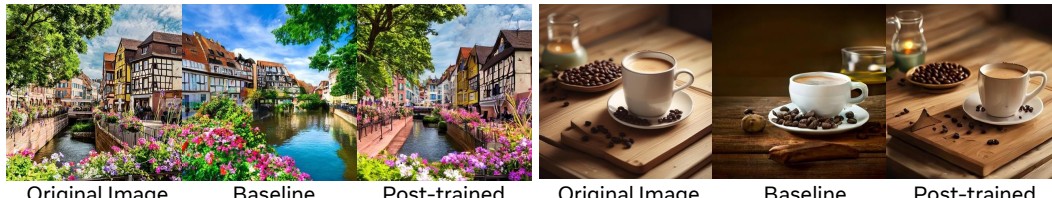

Original Image     Baseline     Post-trained     Original Image     Baseline     Post-trained

Figure 5: **Post-training with RECA restores visual details missed by the baseline models.** For each query image (left), we feed its visual understanding embeddings back into the UMM with the instruction *"Describe the image in detail."* The baseline model (center)'s visual responses, *i.e.*, images, preserve the main subject but distort layout, textures, and colors, while RECA markedly restores visual details like geometry, color, and overall fidelity.

prompt template (e.g., *"Describe the image in detail."*), and ask the UMM to generate, the results in Figure 5 reveal a gap: the main subject is preserved, but spatial layout and composition are scrambled. This raises the hypothesis that the image itself can be treated as a *dense "text prompt"* to train the model, enabling the generation branch of the UMM to better exploit the LLM features and improve its generation capability.

## 2.2 RECA TRAINING PARADIGM

**Training losses.** Traditional UMMs are trained with a combination of text-to-image (T2I) and image-to-text (I2T) objectives. Formally:

$$\mathcal{L}_{\text{t2i}} = \mathcal{L}(f_\theta(t_{\text{prompt}}), I_{\text{gt}}), \quad \mathcal{L}_{\text{i2t}} = \mathcal{L}(f_\theta(\text{concat}(t_{\text{question}}, \mathbf{h}_v)), t_{\text{answer}}) \tag{1}$$

where $\mathcal{L}(\cdot, \cdot)$ denotes the training loss (*e.g.*, cross-entropy for autoregressive models (Xie et al., 2025b; Chen et al., 2025e), diffusion loss for diffusion-based models (Zhou et al., 2025; Deng et al., 2025)), $t_{\text{prompt}}, t_{\text{question}}, t_{\text{answer}}$ are text inputs/outputs, $\mathbf{h}_v$ are embeddings extracted from the understanding encoder, and $I_{\text{gt}}$ is the ground-truth image (we omit the explicit VAE decoder notation for simplicity). $f_\theta(\cdot)$ represents the UMM with parameters $\theta$. Preliminaries of different image generation paradigms are listed in Appendix A.

Our key idea is to replace conventional T2I supervision with a semantic-level image reconstruction loss. Instead of using text captions that are inherently sparse in visual information, we condition the UMM on its own visual understanding embeddings. The reconstruction loss is:

$$\mathcal{L}_{\text{RECA}} = \mathcal{L}(f_\theta(\text{concat}(t_{\text{template}}, \mathbf{h}_v)), I_{\text{gt}}) \tag{2}$$

where $t_{\text{template}}$ is a prompt template triggering image reconstruction (*e.g.*, *Describe the image in detail*). More details of the template collection and visual embedding integration are provided in Appendix C. The overall training loss is:

$$\mathcal{L}_{\text{total}} = \lambda_{\text{RECA}} \mathcal{L}_{\text{RECA}} + \lambda_{\text{i2t}} \mathcal{L}_{\text{i2t}} + \lambda_{\text{t2i}} \mathcal{L}_{\text{t2i}}. \tag{3}$$

We set $\lambda_{\text{RECA}} = 1, \lambda_{\text{t2i}} = 0$. For UMMs that share parameters between understanding and generation, we set $\lambda_{\text{i2t}} = 1$ to preserve I2T capability; otherwise, we set $\lambda_{\text{i2t}} = 0$ and **freeze the understanding component**. We freeze the understanding encoder in most cases, except when the UMM employs the same encoder for both tasks (*e.g.*, Harmon (Wu et al., 2025d)).

**Input resolution.** Previous works show that higher-resolution visual understanding embeddings retain substantially more pixel-level detail (Allakhverdov et al., 2025; Lin et al., 2025). To encourage the model to focus on semantic-level reconstruction, we resize input images to the minimum resolution accepted by the understanding encoder.

**Model inference.** At inference time, our post-trained UMM operates identically to a standard one and requires no additional visual embeddings. For image generation, only a text prompt is needed; for image editing, the inputs remain the text prompt and the original image. This preserves the model's original usability while providing enhanced generation capabilities.

Table 1: **Results on GenEval and DPGBench**. Scores marked with (*) are our reproduced results using 12 random seeds. We use Harmon-1.5B as baseline and post-train it with RECA. The gray-colored rows denote private models, and their results are cited from (Yan et al., 2025b; Geng et al., 2025). Arrows (↑) denote that higher is better. Detailed comparison is listed in Appendix F.3.

| Model | Params | GenEval ↑ | | | | | | | DPG↑ |
| | | Single Obj. | Two Obj. | Counting | Colors | Position | Color Attri. | Overall | Score |
|---|---|---|---|---|---|---|---|---|---|
| Harmon* | 1.5B | 0.99 | 0.87 | 0.69 | 0.86 | 0.45 | 0.51 | 0.73 | 80.93 |
| SD3-Medium | 2B | 0.99 | 0.94 | 0.72 | 0.89 | 0.33 | 0.60 | 0.74 | 84.08 |
| Janus-Pro | 7B | 0.99 | 0.89 | 0.59 | 0.90 | **0.79** | 0.66 | 0.80 | 84.33 |
| FLUX-dev | 12B | 0.99 | 0.85 | 0.74 | 0.79 | 0.21 | 0.48 | 0.68 | 84.00 |
| BAGEL* | 14B | 0.99 | 0.93 | 0.80 | 0.86 | 0.51 | 0.63 | 0.79 | 84.03 |
| Playground-v3 | 24B | 0.99 | 0.95 | 0.72 | 0.82 | 0.50 | 0.54 | 0.76 | 87.06 |
| GPT-4o-Image | - | 0.99 | 0.92 | **0.85** | 0.89 | 0.74 | 0.71 | 0.84 | 86.23 |
| **RECA** | 1.5B | 1.00 | **0.98** | 0.71 | **0.93** | 0.76 | **0.77** | **0.86** | **87.21** |
| *Models Trained with GPT-4o Data* | | | | | | | | | |
| Ovis-U1 | 3.6B | 0.98 | **0.98** | **0.90** | 0.92 | 0.79 | 0.75 | 0.89 | 83.72 |
| OmniGen2 | 7B | 1.00 | 0.95 | 0.64 | 0.88 | 0.55 | 0.76 | 0.80 | 83.57 |
| BLIP3-o* | 8B | 1.00 | 0.92 | 0.63 | 0.91 | 0.86 | 0.67 | 0.83 | 80.73 |
| **RECA** | 1.5B | 1.00 | 0.97 | 0.76 | **0.94** | **0.91** | **0.83** | **0.90** | **88.15** |

## 2.3 DIFFERENCE BETWEEN RECA AND PREVIOUS WORKS

Previous works integrate image reconstruction in different ways: (I) **diffusion-supervised enhancement** (*e.g.*, ViLex) (Wang et al., 2025d; Ma et al., 2025a; Luo et al., 2024; Wang et al., 2024b) leverages pretrained diffusion models to *regularize vision encoders* and improve *visual understanding*; (II) **reconstruction from hidden states** (*e.g.*, ROSS) (Wang et al., 2024a; 2025b)) *adds lightweight decoders* to reconstruct input images from *intermediate embeddings*, thereby regularizing the model to preserve fine-grained details for *visual understanding*; (III) **representation alignment** (*e.g.*, REPA, VA-VAE) (Yu et al., 2024; Yao et al., 2025) introduces *additional alignment modules* that map hidden states in DiT (Peebles & Xie, 2022; Ma et al., 2024a) or VAE (Kingma & Welling, 2013) to representations obtained from *external, pretrained vision encoders*; and (IV) **reconstruction as a prior** (*e.g.*, Lumos) (Ma et al., 2025b) adds *additional DINO features (Caron et al., 2021) into the attention blocks of the diffusion model*, which is further trained on large-scale text-to-image data.

RECA treats images as *dense prompts* and adopts *semantic-level reconstruction* as a native **post-training** objective for UMMs, requiring no auxiliary modules or text-to-image data. RECA is fundamentally different in terms of the methodology, architecture, motivation and task.

## 3 EXPERIMENTS

We validate our RECA method across various unified multimodal models (UMMs), image datasets, and evaluation benchmarks. In particular, we investigate the following aspects:

- **SOTA Results:** RECA achieves state-of-the-art performance on both image generation and editing benchmarks. (Table 1, Table 3)
- **Generality:** RECA delivers consistent performance gains across diverse UMM frameworks, demonstrating its generalizability. (Table 2, Figure 6, Figure 7)
- **Robustness:** RECA consistently improves generation capabilities across diverse datasets and benchmarks, indicating that *the gains are not from training-data memorization*. (Table 5)
- **Training Paradigm:** RECA serves as a post-training method applied after UMM pre-training, and is most effective when used at the final stage of model training. (Table 5, Table 6)

## 3.1 EXPERIMENT SETUP

**Model architectures.** We evaluate RECA across four open-source UMM architectures:

- **Show-o (Discrete)** (Xie et al., 2025b) employs discrete token generation with MaskGIT paradigm (Chang et al., 2022), using CLIP (Radford et al., 2021) / VQGAN (Esser et al., 2021) as the understanding / generation encoder, evaluated at 256×256 and 512×512 resolutions. Our main results adopt the CLIP-understanding variant, while Appendix F.2 explores and analyzes the VQGAN-understanding variant in detail.

Table 2: **RECA consistently improves image generation performance across unified multimodal models**. We report results for the largest model/resolution variant of each architecture; results for smaller models and detailed WISE scores are provided in Appendix F.3.

| Model | RECA | GenEval | | | | | | | DPG | WISE |
|---|---|---|---|---|---|---|---|---|---|---|
| | | Single | Two | Count | Color | Position | Attri. | Overall | | |
| Show-o | ✗ | 97.2 | 80.3 | 61.9 | 78.2 | 27.3 | 52.3 | 66.2 | 82.21 | 0.40 |
| | ✓ | 98.2 (+1.0) | 90.6 (+10.3) | 66.8 (+4.9) | 84.1 (+5.9) | 37.4 (+10.1) | 56.8 (+4.5) | 72.3 (+6.1) | 84.94 (+2.73) | 0.40 (0.00) |
| OpenUni | ✗ | 99.1 | 71.8 | 51.9 | 83.9 | 23.3 | 41.6 | 61.9 | 79.02 | 0.43 |
| | ✓ | 99.1 (0.0) | 92.7 (+20.9) | 52.3 (+0.4) | 87.1 (+3.2) | 43.8 (+20.5) | 70.3 (+28.7) | 74.1 (+12.2) | 82.75 (+3.73) | 0.54 (+0.11) |
| Harmon | ✗ | 99.4 | 87.3 | 68.7 | 86.4 | 44.9 | 51.1 | 72.9 | 80.93 | 0.41 |
| | ✓ | 99.9 (+0.5) | 97.7 (+10.4) | 71.4 (+2.7) | 92.6 (+6.2) | 75.7 (+30.8) | 76.6 (+25.5) | 85.7 (+12.8) | 87.21 (+6.28) | 0.50 (+0.09) |
| BAGEL | ✗ | 99.1 | 93.0 | 79.9 | 86.0 | 51.3 | 63.4 | 78.8 | 84.03 | 0.50 |
| | ✓ | 99.3 (+0.2) | 93.9 (+0.9) | 80.3 (+0.4) | 87.6 (+1.6) | 60.8 (+9.5) | 72.6 (+9.2) | 82.4 (+3.6) | 85.29 (+1.26) | 0.52 (+0.02) |

Table 3: **Results on image editing benchmarks**. We compare RECA post-trained BAGEL with previous methods. All scores are local reproduced. The gray-colored rows denote private models, and their results are cited from (Deng et al., 2025; Ye et al., 2025).

| Method | ImgEdit | | | | | | | | | | GEdit-Bench-EN | | |
|---|---|---|---|---|---|---|---|---|---|---|---|---|---|
| | Bg. | Style | Adj. | Ext. | Rem. | Rep. | Add | Comp. | Act. | Ovr. | SC | PQ | Overall |
| Gemini 2.0 | - | - | - | - | - | - | - | - | - | - | 5.43 | 6.78 | 5.36 |
| GPT-4o-Image | 4.57 | 4.93 | 4.33 | 2.90 | 3.66 | 4.35 | 4.61 | 3.96 | 4.89 | 4.20 | 7.85 | 7.62 | 7.53 |
| BAGEL-NHR | 3.56 | 4.43 | 3.62 | 1.57 | 3.17 | 3.98 | 4.12 | 2.88 | 3.85 | 3.48 | 8.04 | 6.87 | 7.08 |
| FLUX-Kontext | **3.89** | 4.62 | 3.69 | **1.81** | 2.97 | 4.20 | 3.80 | **3.00** | 4.18 | 3.60 | 6.95 | **7.30** | 6.27 |
| BAGEL | 3.38 | 4.53 | 3.58 | 1.49 | 3.15 | 3.82 | 3.71 | 2.64 | 4.21 | 3.38 | 7.96 | 6.64 | 6.94 |
| **BAGEL-RECA** | 3.85 | **4.73** | **3.86** | 1.68 | **3.75** | **4.28** | **4.20** | 2.94 | **4.56** | **3.75** | **8.24** | 6.87 | **7.27** |
| *vs. baseline* | +0.47 | +0.20 | +0.28 | +0.19 | +0.60 | +0.46 | +0.49 | +0.30 | +0.35 | +0.37 | +0.28 | +0.23 | +0.33 |

- **Harmon (MAR)** (Wu et al., 2025d) adopts masked autoregressive generation, with MAE (Li et al., 2024) / MAE & VAE (Kingma & Welling, 2013) as the understanding / generation encoder, evaluated at 0.5B and 1.5B parameter scales.
- **OpenUni (Continuous)** (Wu et al., 2025c) generates in continuous latent space via diffusion, with InternVL3 (Zhu et al., 2025) / VAE (Kingma & Welling, 2013) as the understanding / generation encoder. It serves as an open-source counterpart of MetaQueries (Pan et al., 2025b); we evaluate both 1.6B and 3.6B variants at 512×512 resolution without GPT-4o-Image distillation.
- **BAGEL (Continuous)** (Deng et al., 2025) also employs continuous diffusion generation, with SigLIP (Zhai et al., 2023) / VAE (Kingma & Welling, 2013) as the understanding / generation encoder. We report the results of BAGEL at 1024×1024 resolution.

Together, these models cover the main families of UMM generation mechanisms: discrete token prediction, masked autoregressive (MAR), and continuous diffusion.

**Evaluation details.** We evaluate text-to-image generation on GenEval (Ghosh et al., 2023) and DPGBench (Hu et al., 2024), and image editing on ImgEdit (Ye et al., 2025) and GEdit-Bench-EN (Liu et al., 2025). Our baselines include both generation-only models (e.g., SDXL (Podell et al., 2023), DALL-E 3 (Betker et al., 2023)) and unified multimodal models (e.g., Show-o (Xie et al., 2025b), Harmon (Wu et al., 2025d), BAGEL (Deng et al., 2025), GPT-4o-Image (OpenAI, 2024)).

**Training data.** We post-train UMMs with high-quality open-source data, including MidjourneyV6 (CortexLM, 2024), LLaVA Mix-665K (Liu et al., 2024b), and 10,000 FLUX-generated samples (jackyhate, 2024). Further implementation, evaluation and dataset details are provided in Appendix E. To ensure fair comparison, our main experiments **exclude GPT-4o-Image distillation data like BLIP3o-60k**, which can inflate benchmark scores due to *GenEval Template Leakage*. See Appendix F.1 for further discussion.

## 3.2 RECA ACHIEVES SOTA PERFORMANCE ON IMAGE GENERATION AND EDITING

**SOTA performance without GPT-4o-Image distillation.** As shown in Table 1, post-training the Harmon-1.5B model with RECA achieves SOTA overall scores on GenEval and DPGBench. Despite using significantly fewer parameters, RECA surpasses recent methods, *e.g.*, Janus-Pro (Chen et al., 2025d) and BAGEL (Deng et al., 2025).

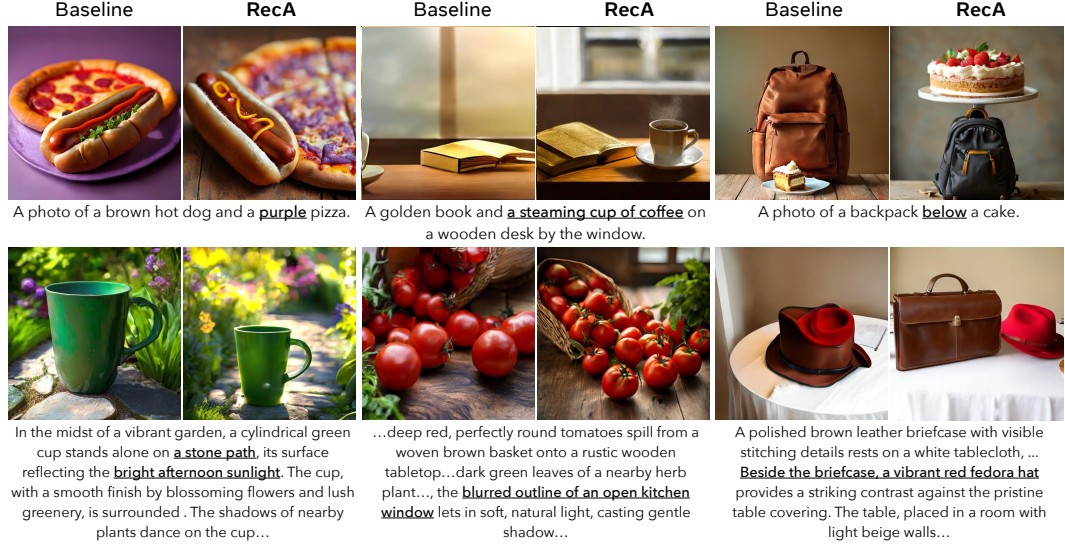

Figure 6: **Image generation results *vs.* baselines.** We use Harmon-1.5B as baseline. The post-trained model better handles multiple objects, complex attributes and spatial layouts, preserving fine details missed by the baseline.

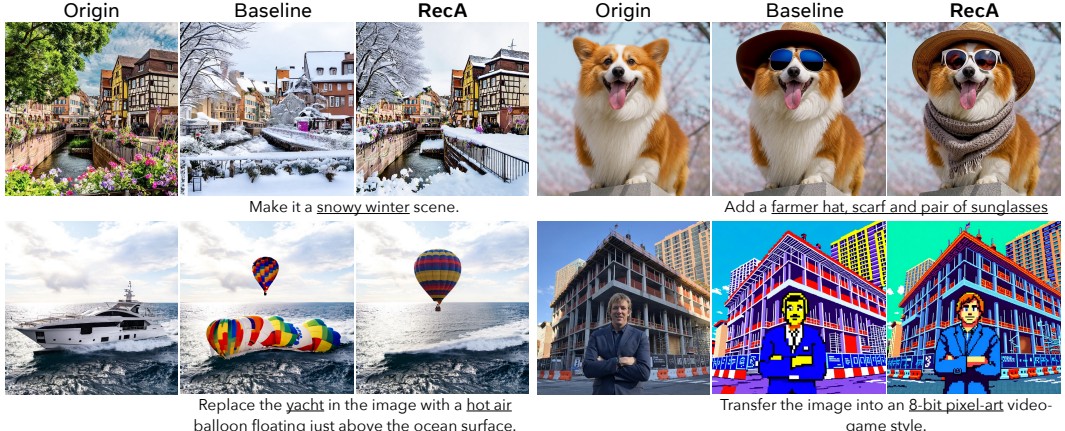

Figure 7: **Image editing results *vs.* baselines**. We use BAGEL as baseline. Our model consistently improves performance on object addition, replacement, style transfer, and scene modification.

**Image editing results are also substantially improved**. As shown in Table 3, RECA consistently outperforms existing baselines across all subtasks and benchmarks. On ImgEdit, RECA achieves an overall score of 3.75, surpassing FLUX-Kontext (3.60) and the baseline (3.38). On GEdit-Bench-EN, RECA reaches 7.27, which closes the gap between open-source models and GPT-4o-Image. In particular, with only 10,000 unlabeled images and 27 GPU hours, RECA beats the concurrent work BAGEL-NHR (Kuprashevich et al., 2025) (3.48), which employs supervised fine-tuning (SFT) on 300,000 high-quality image editing datat.

## 3.3 RECA IS EFFECTIVE ACROSS DIFFERENT UMM FRAMEWORKS

**Consistent performance gains across UMM architectures.** As shown in Table 2, RECA demonstrates consistent and significant improvements across all evaluated UMM frameworks. We report results for the largest model variant of each architecture, with smaller model results provided in Appendix F.3. On GenEval and DPGBench, the most substantial improvement is achieved by Harmon-1.5B, which attains scores of 85.7 (+12.8) and 87.21 (+6.28), respectively.

**Performance on WISE benchmark.** WISE (Niu et al., 2025b) evaluates text-to-image reasoning with 1,000 knowledge-puzzle prompts (*e.g.*, Einstein's favorite instrument). As shown in Table 2,

| Dataset | SFT | RECA | GenEval | DPG |
|---------|-----|------|---------|-----|
| MidjourneyV6 | ✓ | ✗ | 74.76 | 80.89 |
| MidjourneyV6 (dense) | ✓ | ✗ | 74.05 | 80.67 |
| MidjourneyV6 | ✗ | ✓ | **85.69** | **87.21** |
| MidjourneyV6 | ✓ | ✓ | 81.22 | 86.54 |
| BLIP3o-60k | ✓ | ✗ | 84.95 | 85.19 |
| BLIP3o-60k | ✗ | ✓ | 85.21 | **86.50** |
| BLIP3o-60k | ✓ | ✓ | **86.14** | 85.96 |
| BLIP3o-60k (w/o GenEval) | ✓ | ✗ | 80.88 | 85.24 |
| BLIP3o-60k (w/o GenEval) | ✗ | ✓ | **84.76** | **86.37** |

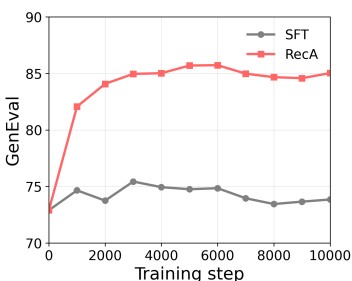

Table 5: **Comparison of SFT and RECA across post-training datasets.** RECA consistently outperforms SFT, especially when *template leakage data* is excluded (**yellow row**), demonstrating that the improvements are genuine and robust. All results are reported at 5k steps. **Right:** GenEval overall scores of Harmon-1.5B with SFT *vs.* RECA across training steps.

RECA improves Harmon and OpenUni, while BAGEL and Show-o show limited or no gains. We conclude that RECA primarily improves semantic-level alignment, while reasoning ability remains an open problem for future work. Detailed WISE results are provided in Appendix F.3.

## 3.4 MORE RESULTS

**Qualitative results.** For image generation, the baseline often fails on *multiple objects*, *complex attributes*, and *spatial layouts*, and in Figure 6, the post-trained model preserves these fine-grained details well (e.g., kitchen window, sunlight). For image editing, Figure 7 shows consistent improvements across tasks such as object addition, replacement, style transfer, and color modification. Additional examples are provided in Appendix I and J.

**Visual understanding results.** We investigate the effect of RECA on UMMs that share parameters between understanding and generation (e.g., Harmon-1.5B). In Table 4, results on POPE (Li et al., 2023b), MME (Fu et al., 2023), GQA (Hudson & Manning, 2019), MMMU (Yue et al., 2024), and SEED (Li et al., 2023a) show that RECA can improve UMMs' generation fidelity without degrading visual understanding. For architectures with decoupled understanding components (OpenUni, BAGEL), their capability remains unchanged.

Table 4: Visual understanding performance.

| Model | MME | POPE (Acc) | POPE (F1) | GQA | MMMU | SEED |
|-------|-----|------------|-----------|-----|------|------|
| Harmon | 1195 | 83.8 | **83.9** | **58.8** | 34.7 | 65.2 |
| RECA | **1223** | 83.9 | 83.2 | 58.4 | **35.7** | **65.3** |

## 3.5 EMPIRICAL STUDIES OF RECA AS A POST-TRAINING METHOD

**SFT vs. RECA as post-training methods.** We directly compare supervised fine-tuning (SFT) with RECA as alternative post-training strategies for UMMs. Unlike SFT, which depends on text–image pairs, RECA is self-supervised and requires only unlabeled images. Our key questions are: *(i) which method is more effective, and (ii) when and how should RECA be applied relative to SFT?*

RECA *demonstrates genuine effectiveness compared with SFT.* As shown in Table 5, on MidjourneyV6, SFT achieves 74.76 on GenEval. Adding denser captions via Midjourney-LLaVA (brivangl, 2024) does not help; GenEval remains nearly unchanged and DPGBench even drops a little. In contrast, replacing SFT with RECA dramatically

Table 6: **Training recipe.** SFT *vs.* RECA as sequential post-training stages.

| RECA Dataset | Order | GenEval | DPG |
|--------------|-------|---------|-----|
| BLIP3o-60k | RECA → SFT | 85.91 | 85.67 |
| BLIP3o-60k | SFT → RECA | 89.00 | 87.50 |
| MidjourneyV6 | SFT → RECA | **90.15** | **88.15** |

boosts performance under the same setup (85.69/87.21). On the complete BLIP3o-60k dataset, RECA still outperforms SFT on both benchmarks. Notably, removing *template leakage data* leads to a performance drop for SFT (-4.07 on GenEval), whereas RECA maintains strong performance. The training dynamics in the right panel of Table 5 further demonstrate that RECA consistently outperforms SFT. While SFT performance stagnates, RECA rapidly ascends to a superior level and maintains a stable advantage throughout the training process; the evolution of GenEval subtasks is provided in Appendix F.5.

*When to apply RECA?* We also study training order to understand when to apply RECA relative to SFT. As shown in Table 5, mixed training (half SFT, half RECA per batch) is unstable: it slightly

helps on BLIP3o-60k but fails on MidjourneyV6. If we do SFT on BLIP3o-60k, sequential training is more decisive: as shown in Table 6, applying RECA after SFT (SFT→RECA) consistently delivers the best results (90.15/88.15 on MidjourneyV6). Reversing the order (RECA →SFT) degrades performance (-3.09/-1.83). This asymmetry underscores that SFT provides broad text–image alignment, while RECA serves best as a refinement stage enhancing semantic grounding and visual faithfulness.

**Visual understanding *vs.* visual generation encoder.** Most UMMs employ two encoders: one for understanding (semantic features) and one for generation (pixel-level features). BAGEL follows this design, making it a natural testbed for RECA. As shown in Table 7, conditioning on the generation encoder (VAE) yields only marginal or degraded gains, while embeddings from the understanding encoder (ViT) produce consistently better results across GenEval, DPGBench, ImgEdit, and GEdit. This indicates that RECA benefits most from semantic embeddings that capture high-level conceptual information, rather than raw visual details.

**Different resolutions for the understanding encoder.** For Show-o, Harmon and OpenUni, the input resolutions are fixed at $336 \times 336$, $512 \times 512$ and $448 \times 448$, respectively. BAGEL uses NaViT (Dehghani et al., 2023) and can flexibly accommodate arbitrary input resolutions. We perform experiments on $224 \times 224$ and $512 \times 512$ input

Table 7: **Vision encoder type and input resolution comparison**. Embeddings from visual understanding encoder with the lowest resolution are the best for RECA.

| Vision Encoder | Resolution | GenEval | DPG | ImgEdit | GEdit |
|---|---|---|---|---|---|
| Baseline | - | 78.8 | 84.03 | 3.38 | 6.94 |
| VAE (gen.) | $256 \times 256$ | 78.5 | 83.92 | 3.63 | 7.08 |
| ViT (und.) | $224 \times 224$ | **82.4** | **85.29** | **3.75** | **7.27** |
| ViT (und.) | $512 \times 512$ | 79.2 | 84.61 | 3.68 | 7.18 |

resolutions for RECA, and the results in Table 7 show that the model post-trained on $224 \times 224$ consistently outperforms its $512 \times 512$ counterpart across all benchmarks.Prior work shows that higher-resolution embeddings retain substantially more pixel-level detail (Allakhverdov et al., 2025; Lin et al., 2025), which in turn drives the model to lean on low-level cues rather than the semantic features that RECA is designed to emphasize.

**Conclusion.** Across data and strategies, RECA consistently proves to be a stronger post-training method than SFT. **The best post-training recipe is a two-stage pipeline**: first SFT on high-quality paired data for coarse alignment, followed by RECA for self-supervised fine-grained refinement.

## 4 RELATED WORK

### 4.1 UNIFIED MULTIMODAL MODELS (UMMs)

Unified multimodal model (UMM) is a backbone capable of multimodal understanding and generation within a single model or system (Zhang et al., 2025b). Recent studies have explored different types of UMMs: **(I) Discrete UMM**. Models like Chameleon tokenize images and predict them autoregressively in a next-token prediction way (Team, 2024; Qu et al., 2024; Chen et al., 2025e; Wu et al., 2024a), while Show-o (Xie et al., 2025b) introduces a *discrete-diffusion* schedule to improve the token prediction process. **(II) Continuous UMM**. Models like Transfusion attach a diffusion (or flow-matching) (Ho et al., 2020; Lipman et al., 2023) head to the shared transformer (Zhou et al., 2025; Ma et al., 2024b), while some UMMs keep pretrained MLLMs frozen and route its intermediate features with learnable queries to an external image generator (Pan et al., 2025b; Lin et al., 2025). **(III) MAR UMM**. Masked-autoregressive (MAR) (Li et al., 2024) is a novel autoregressive image generation paradigm, which has been adopted by models like Harmon (Wu et al., 2025d; Fan et al., 2025a; Yang et al., 2024b; Wang et al., 2025c).

### 4.2 POST-TRAINING STRATEGIES FOR UMMs

Recent efforts explore different post-training techniques for enhancing the generation capability of UMMs. **(I) Chain-of-Thought (Wei et al., 2022) or test-time verification** do reasoning before the generation or verify generated image step-by-step, but they depend on external models and do not improve the UMM's native generation capability. (Guo et al., 2025; Wang et al., 2025e; Fang et al., 2025; Tian et al., 2025). **(II) Reinforcement learning.** Methods such as DPO (Rafailov et al., 2023) and GRPO (Shao et al., 2024) optimize generation policies with human or automatic preference signals, but require curated paired data and carefully tuned advantage functions. (Wei

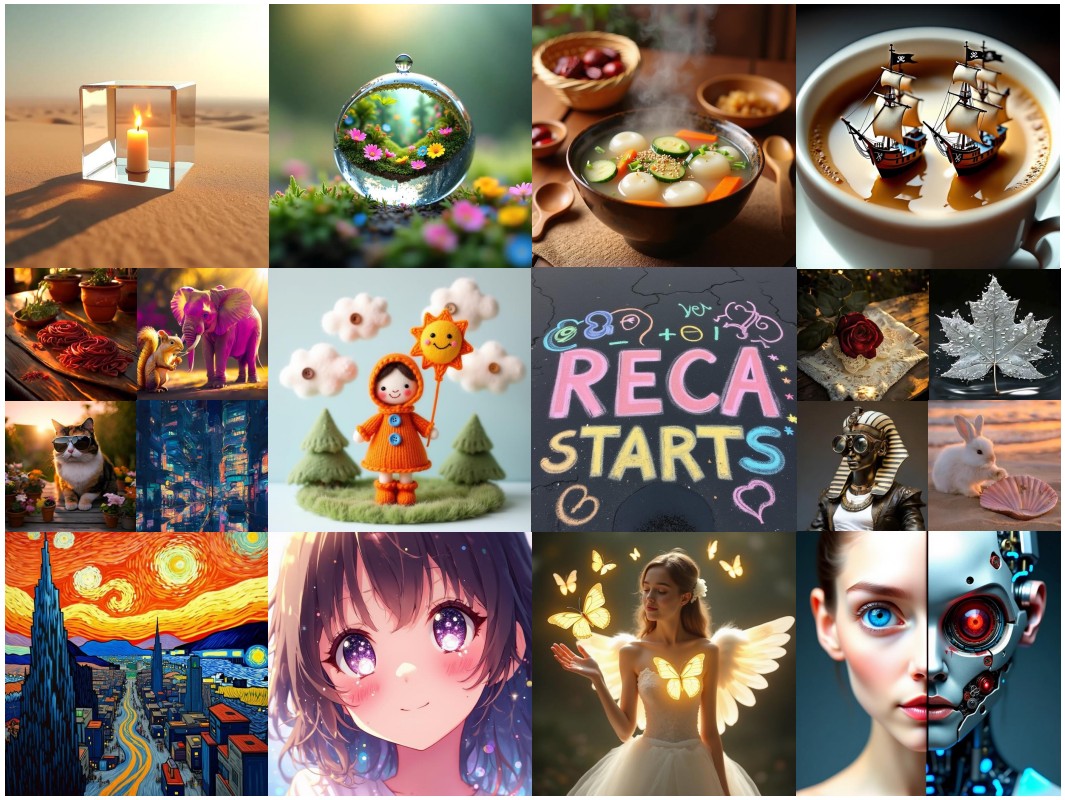

Figure 8: **Qualitative T2I results**. The large images ($1024 \times 1024$) are generated by the post-trained BAGEL, while the small images ($512 \times 512$) are generated by the post-trained Harmon. Detailed captions are listed in Appendix H.

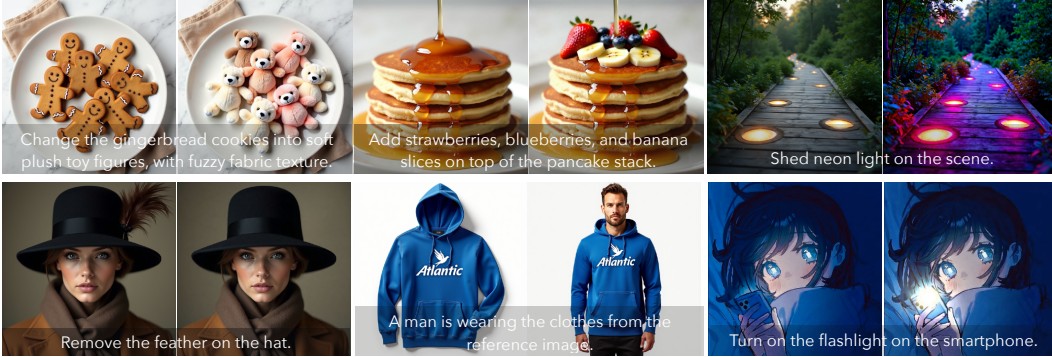

Figure 9: **Qualitative image editing results.** Edited outputs are generated by BAGEL post-trained with RECA. Left: original images; Right: edited images.

et al., 2025; Han et al., 2025; Yan et al., 2025a; Mao et al., 2025; Tian et al., 2025; Jiang et al., 2025).
**(III) High-quality synthetic data.** Recent works (Chen et al., 2025b; Wang et al., 2025f) construct large-scale GPT-4o–Image (OpenAI, 2024) generated image–text pairs for supervised fine-tuning, which boost benchmarks but incur high data-generation cost and potential distribution shift.

## 5  CONCLUSION

In this work, we propose **RECA**, a lightweight post-training method that replaces sparse text-to-image supervision with dense features from the model's own visual understanding encoder. RECA requires no extra caption data yet significantly improves image generation and editing across architectures. We discuss the limitations and future directions in Appendix G.

REPRODUCIBILITY STATEMENT

We ensure reproducibility by providing all training and evaluation hyperparameters, along with full implementation details, in Appendix E. These materials are sufficient to replicate our experiments. Detailed prompt templates are included in the Supplementary Material. In line with our commitment to open-source research, we will make our codes and model weights publicly available.

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

We provide additional information in the supplementary material, as outlined below:

- **Sec. A**: Brief introduction to different image generation paradigms.
- **Sec. B**: Difference Between RECA and Classifier-Free Guidance (CFG).
- **Sec. C**: Prompt templates for RECA.
- **Sec. D**: Motivation and Mechanism Analysis of RECA.
- **Sec. E**: Further implementation details for RECA implementation and evaluation.
- **Sec. F**: Additional experimental results and analysis.
- **Sec. G**: Limitations and future work for RECA.
- **Sec. H**: Generated captions for figures in the DEMO.
- **Sec. I**: Additional qualitative results on image editing tasks.
- **Sec. J**: More uncurated qualitative examples for text-to-image generation.
- **Sec. K**: The Use of Large Language Models (LLMs).

# A  PRELIMINARIES OF IMAGE GENERATION PARADIGMS

## A.1  DISCRETE PARADIGM

Discrete paradigms use image tokenizer (Van Den Oord et al., 2017; Esser et al., 2021; Chang et al., 2022) and treat images as a sequence of discrete tokens, predicting each token in turn. The joint likelihood factorizes as:

$$p(x) = \prod_{i=1}^{N} p_\theta(x_i \mid \mathbf{x}_{<i}),$$

$$\mathcal{L}_{\text{AR}} = -\mathbb{E}_{x \sim p_{\text{data}}} \sum_{i=1}^{N} \log p_\theta(x_i \mid \mathbf{x}_{<i}), \quad (4)$$

where the model is trained by minimizing the cross-entropy of each next-token prediction. At sampling time, tokens are drawn sequentially ($x_1 \sim p_\theta(x_1)$, $x_2 \sim p_\theta(x_2 \mid x_1)$, etc.). Models such as Chameleon (Team, 2024) and Janus (Chen et al., 2025e) append it onto LLM's LM head.

MaskGIT (Chang et al., 2022) extends the discrete paradigm by parallel prediction with random masks. Given a tokenized image sequence $\mathbf{y} = (y_1, \ldots, y_N)$, where $N$ is the number of discrete tokens obtained from a image tokenizer's codebook of size $K$, and a binary mask vector $\mathbf{m} = (m_1, \ldots, m_N)$ with $m_i = 1$ indicating that position $i$ is masked, the model is trained with the masked prediction loss:

$$\mathcal{L}_{\text{mask}} = -\mathbb{E}_{\mathbf{y},\mathbf{m}} \sum_{i=1}^{N} m_i \log p_\theta(y_i \mid \mathbf{y}^M) \quad (5)$$

where $\mathbf{y}^M$ denotes the input sequence with masked tokens replaced by a special `[MASK]` symbol. During inference, the model starts from a fully masked input and iteratively fills it with predicted image tokens. At step $t$, the model predicts distributions $p_i^t = p_\theta(y_i \mid \mathbf{y}^{M,t})$ for all currently masked positions, where $p_i^{(t)}$ is a categorical distribution over the $K$ image tokens. The confidence score is defined as $c_i^t = \max_j p_{ij}^t$, i.e., the maximum probability across categories. A scheduling function $\gamma(t/T)$, where $T$ is the total number of refinement steps, controls the number of tokens to remain masked:

$$n_t = \lfloor N \cdot \gamma(t/T) \rfloor,$$

$$m_i^{(t+1)} = \mathbb{1}\left[c_i^{(t)} \leq \text{the } n_t\text{-th smallest confidence}\right]. \quad (6)$$

This process progressively decreases the mask ratio until the full image token sequence is generated. In contrast to autoregressive next-token prediction, MaskGIT decodes multiple tokens in parallel with bidirectional context, achieving faster sampling and higher generation fidelity. UMMs such as Show-o (Xie et al., 2025b) use this paradigm.

## A.2 CONTINUOUS PARADIGM

The most representative method in the continuous paradigm is diffusion (Sohl-Dickstein et al., 2015; Ho et al., 2020), which learns a gradual noising process from Gaussian distribution. The forward (diffusion) process progressively corrupts a clean image $\mathbf{x}_0 \sim p_{\text{data}}$ into pure Gaussian noise $\mathbf{x}_T$:

$$
\begin{aligned}
q(\mathbf{x}_t \mid \mathbf{x}_{t-1}) &= \mathcal{N}\big(\mathbf{x}_t; \sqrt{1 - \beta_t}\, \mathbf{x}_{t-1},\, \beta_t \mathbf{I}\big), \\
q(\mathbf{x}_t \mid \mathbf{x}_0) &= \mathcal{N}\big(\mathbf{x}_t; \sqrt{\bar{\alpha}_t}\, \mathbf{x}_0,\, (1 - \bar{\alpha}_t)\mathbf{I}\big),
\end{aligned}
\tag{7}
$$

where $\beta_t \in (0, 1)$ are pre-defined hyperparamters, $\alpha_t = 1 - \beta_t$, and $\bar{\alpha}_t = \prod_{s=1}^{t} \alpha_s$. The model $\theta$ learns the reverse denoising distribution:

$$
p_\theta(\mathbf{x}_{t-1} \mid \mathbf{x}_t) = \mathcal{N}\big(\mathbf{x}_{t-1}; \mu_\theta(\mathbf{x}_t, t, z),\, \sigma_t^2 \mathbf{I}\big),
\tag{8}
$$

and is trained with the noise prediction loss:

$$
\begin{aligned}
\mathcal{L}_{\text{diff}} &= \mathbb{E}_{\mathbf{x}_0, \epsilon, t, z}\Big[\|\epsilon - \epsilon_\theta(\mathbf{x}_t, t, z)\|^2\Big], \\
\mathbf{x}_t &= \sqrt{\bar{\alpha}_t}\, \mathbf{x}_0 + \sqrt{1 - \bar{\alpha}_t}\, \epsilon, \quad \epsilon \sim \mathcal{N}(0, \mathbf{I}).
\end{aligned}
\tag{9}
$$

Here $\epsilon_\theta(\mathbf{x}_t, t, z)$ is the predicted noise given the noisy input $\mathbf{x}_t$, timestep $t$ and condition signal $z$. Generation starts from $\mathbf{x}_T \sim \mathcal{N}(0, \mathbf{I})$ and applies the learned reverse process until $\mathbf{x}_0$ is recovered.

Instead of discrete noising steps, flow-matching (Lipman et al., 2023) defines a continuous trajectory $\{\mathbf{x}_t\}_{t \in [0,1]}$ that transports a prior distribution $p_0$ (e.g. Gaussian noise at $t = 0$) into the data distribution $p_1$ ($t = 1$). In ordinary differential equation (probability flow ODE):

$$
\frac{\mathrm{d}}{\mathrm{d}t}\mathbf{x}(t) = u_\theta(\mathbf{x}(t), t, z),
\tag{10}
$$

where $u_\theta$ is a neural velocity (vector) field. The ground-truth target field $u^*(x, t, z)$ can be derived from an optimal coupling between $p_0$ and $p_1$, and training minimizes:

$$
\mathcal{L}_{\text{flow}} = \mathbb{E}_{\mathbf{x}, t, z}\big[\|u_\theta(\mathbf{x}, t, z) - u^*(\mathbf{x}, t, z)\|^2\big].
\tag{11}
$$

Rectified flow (Liu et al., 2022) simplifies this by defining a linear interpolation:

$$
\mathbf{x}(t) = (1 - t)\mathbf{x}(0) + t\epsilon, \quad \epsilon \sim \mathcal{N}(0, \mathbf{I}),
\tag{12}
$$

so that the target velocity field is available in closed form:

$$
u^*(\mathbf{x}(t), t, z) = \frac{\mathrm{d}}{\mathrm{d}t}\mathbf{x}(t) = \epsilon - \mathbf{x}(0).
\tag{13}
$$

Diffusion adds Gaussian noise at each step with variance $\beta_t$, whereas rectified flow directly defines a continuous interpolation between data and noise. UMMs such as BAGEL (Zhou et al., 2025), Janusflow (Ma et al., 2024b) and Metaqueries (Pan et al., 2025b; Wu et al., 2025c) append these paradigms above onto LLM.

**UMM Input**

*<|im_start|>system\nYou are a helpful assistant.<|im_end|>\n<|im_start|>user\n*

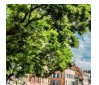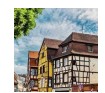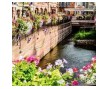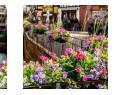

*Describe this image in detail.<|im_end|>\n<|im_start|>assistant*

**Supervised Output**

Image captioning: *This image depicts a beautiful town and a canal, surrounded by colorful flowers…*

Reconstruction Alignment:

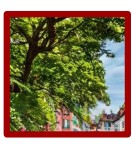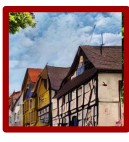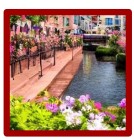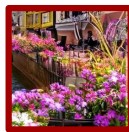

Figure 10: **Visual embedding integration in RECA.** The left side shows the UMM input format, where visual understanding embeddings (represented as image patches) are concatenated with text prompts following standard image captioning format. The right side contrasts two supervision paradigms: image captioning uses sparse text descriptions as supervision, while RECA uses the original images themselves as dense reconstruction targets.

### A.3 MASKED AUTOREGRESSIVE (MAR)

MAR (Li et al., 2024) is a mask-based generation paradigm similar to MaskGIT, but it generalizes AR without vector quantization. Specifically, MAR employs a diffusion decoder to generate images with continuous tokens in the transformer.

Inference also differs from MaskGIT: **(I)** MAR predicts continuous features instead of logits over discrete codes, and **(II)** while MaskGIT fills the most confident logits, MAR stochastically samples denoised patches at each step. UMMs such as Fluid (Fan et al., 2025b) and Harmon (Wu et al., 2025d) follow this design.

## B DIFFERENCE BETWEEN RECA AND CLASSIFIER-FREE GUIDANCE (CFG)

**Classifier-free guidance** (CFG (Ho & Salimans, 2022)) is typically used to improve image generation fidelity. At each generation step, we compute a conditional prediction $\mathbf{o}_{\text{cond}}$ and an unconditional prediction $\mathbf{o}_{\text{uncond}}$, where $\mathbf{o}$ denotes either the autoregressive head's logits or the diffusion head's predicted noise, as we introduced in Sec. A. The output is given by:

$$\mathbf{o} = (1 + \omega)\,\mathbf{o}_{\text{cond}} - \omega\,\mathbf{o}_{\text{uncond}}, \tag{14}$$

with $\omega$ the guidance scale. RECA is conceptually orthogonal to CFG. Whereas CFG relies on contrast between a conditional and a null-text (or template) prompt, RECA leverages embeddings from the visual understanding encoder as dense prompts for reconstruction-based alignment. The two techniques are fully compatible and can be applied together.

## C PROMPT TEMPLATES FOR RECA

### C.1 VISUAL EMBEDDING INTEGRATION

The integration of visual understanding embeddings in RECA follows the standard multimodal input paradigm used in modern MLLMs for image captioning tasks. As illustrated in Figure 10, the input sequence consists of:

1. Standard instruction tuning template (e.g., `<|im_start|>system\nYou are a helpful assistant.<|im_end|>`).
2. Tokenized image representations from the visual understanding encoder (e.g., CLIP, SigLIP).
3. A template triggering image description or reconstruction.
4. Trigger (e.g., `<|im_start|>assistant`).

During RECA training, the model is conditioned on the concatenation of the text template and visual embeddings, and trained to reconstruct the original image. This process is identical to how MLLMs

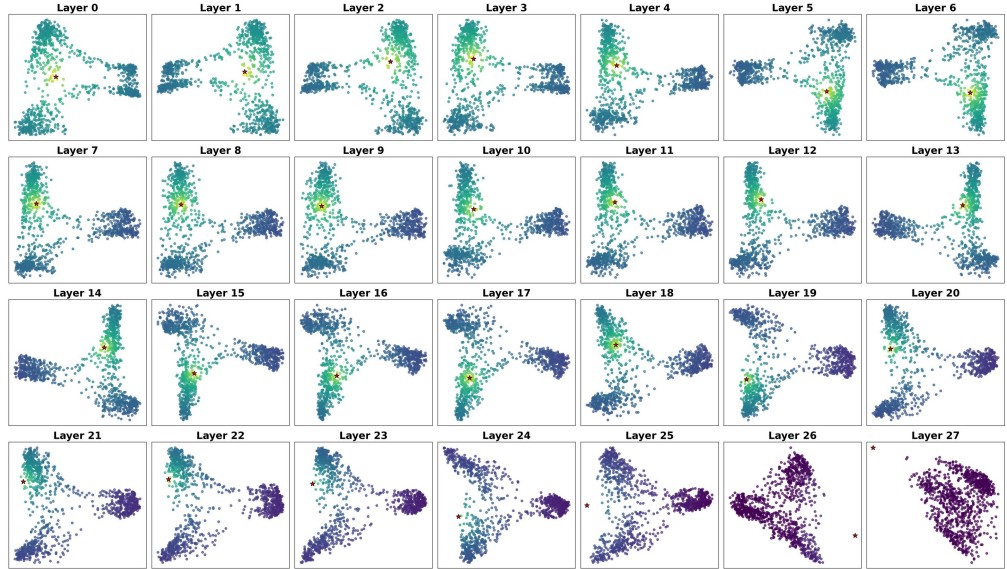

Figure 11: **Deep Feature Alignment.** PCA (Abdi & Williams, 2010) visualization of text embeddings (red mark, the word "black") and image embeddings (a photo of a black dog) across model layers. In deep transformer layers, text and visual features share the same semantic manifold. This indicates that visual embeddings effectively function as dense, language-aligned text prompts.

process images for captioning tasks, except the supervision signal comes from image reconstruction loss rather than text generation loss.

## C.2 Prompt Template Diversification

For template diversification in RECA post-training, we utilize GPT-o3 to generate 360 distinct prompt templates based on the seed template "Describe the image in detail." This diversification prevents the model from overfitting to a specific prompt format and ensures robust performance across various instruction phrasings. Below are representative examples from our template collection:

- "Provide a detailed description of the image."
- "What can you observe in this image? Please describe it comprehensively."
- "Analyze and describe the visual elements present in this image."
- "Give a thorough description of what you see in the image."
- "Examine the image carefully and provide a detailed account."
- "Describe the contents of this image in detail."
- "What does this image show? Please provide a comprehensive description."
- "Offer a detailed visual analysis of the image."
- "Describe all the visual elements you can identify in this image."
- "Provide an in-depth description of what is depicted in the image."

The complete collection of 360 templates varies in linguistic structure, complexity, and prompting style while maintaining the core objective of triggering detailed image descriptions.

## D  Mechanistic Analysis of RecA

In this section, we provide a deeper analysis of the mechanism behind semantic reconstruction alignment (RECA). We investigate how the model processes visual information as dense prompts and the correlation between reconstruction and generation capabilities.

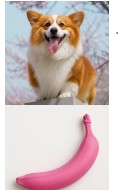 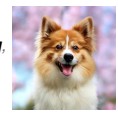 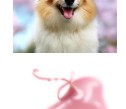 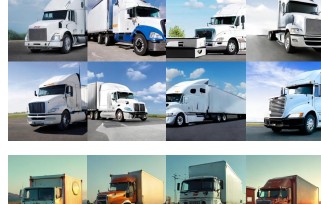

*The image captures a **Shiba Inu dog**, known for its **fox-like** appearance and spirited personality…*

*The image presents a single, vibrant pink **shoe** against a stark white background…*

Input Image      Generated Caption      Reconstructed Image

Dense Image Prompt

*A photo of a truck and a microwave*

Sparse Text Prompt

(a) **Reconstruction reflects internal semantic projection. Top:** The UMM extracts semantic features (e.g., "fox-like") from the input Corgi and resamples a visually consistent image. **Bottom:** For an anomalous concept (pink banana), the UMM maps the visual input to the semantic concept of a "pink shoe" and then faithfully produces a shoe. This confirms that generation is driven by the model's internal semantic interpretation.

(b) **UMM's Reconstruction correlates with generation capability. Top:** Provided with the dense image prompt, the UMM consistently fails to reconstruct the microwave across multiple seeds. **Bottom:** Consistent with the reconstruction failure, the text prompt also fails to generate the microwave. This suggests a strong correlation between the two tasks.

**Visual-Textual Feature Isomorphism.** A fundamental premise of RECA is that visual embeddings from the understanding encoder can serve as dense supervision for the generator. As shown in Figure 11, in the deeper layers of the transformer (Layers $3 \sim 20$), text features (e.g., the word "black") and image features (e.g., a photo of a black dog) aligns. This suggests that visual and textual information is effectively isomorphic in the semantic space, allowing us to use images as dense prompts to supervise the generation backbone.

**Internal Semantic Projection.** As shown in Figure 12a, We observe that the UMM treats input images as semantic prompts rather than raw pixels. For standard inputs (e.g., a dog), it performs semantic resampling. Notably, for rare inputs like a *pink banana*, the UMM projects the visual features onto its nearest learned semantic concept, in this case, a "pink shoe". Crucially, it faithfully produces a shoe based on this internal mapping. This confirms that **images are interpreted as text-like semantic concepts**, and the UMM acts upon this interpretation.

**Correlation Between Reconstruction and Generation.** Furthermore, we find a strong empirical correlation between reconstruction and generation. As shown in Figure 12b, we test a challenging composition ("microwave and truck"). (I) When the UMM is conditioned on the **dense image prompt**, it persistently fails to generate the microwave across multiple random seeds. (II) Correspondingly, when conditioned on the **sparse text prompt** (T2I), it also fails to generate.

This suggests that if the model cannot resolve a concept given dense visual cues, it is unlikely to succeed with sparse textual cues. Therefore, enhancing the model's ability to reconstruct complex details serves as a *direct pathway* to improving its generation capabilities. RECA works by refining this mapping process.

# E  EXPERIMENTAL SETUP

**Implementation Details.** We post-train all models on a single NVIDIA A100 (80 GB) GPU. Table 8 presents the detailed hyper-parameter configurations for all models, including Show-o ($256^2/512^2$), Harmon (0.5B/1.5B), OpenUni (1.6B/3.6B), and BAGEL (14B). The table includes optimization settings, trainable modules, and loss weights used during RECA post-training. Hyper-parameters not listed in Table 8 follow the original open-source baselines. All $\lambda_{t2i}$ is set to 0.

For experiments that report averaged metrics (rows marked with $^*$ in Table 1), we run 12 repetitions using random seeds $\{0, 1, \ldots, 11\}$; other experiments default to seed 0 unless otherwise stated. At inference time, we execute Show-o, Harmon, and OpenUni on an NVIDIA RTX 4090 (24 GB), while BAGEL inference remains on the A100.

**Evaluation Details.** Following previous work, we evaluate text-to-image generation capabilities using GenEval (Ghosh et al., 2023), DPGBench (Hu et al., 2024) and WISE (Niu et al., 2025b). Our baselines include: (I) *Generation-only models*: SD3-Medium (Esser et al., 2024), SDXL (Podell et al., 2023), SANA-1.5 (Xie et al., 2025a), Emu3-Gen (Wang et al., 2024c), FLUX-dev (Black-

Table 8: Fine-tuning hyperparameter setup.

| | Show-o | | Harmon | | OpenUni | | BAGEL |
|---|---|---|---|---|---|---|---|
| | $256^2$ | $512^2$ | 0.5B | 1.5B | 1.6B | 3.6B | 14B |
| **Optimization** | | | | | | | |
| Optimizer | AdamW | AdamW | AdamW | AdamW | AdamW | AdamW | AdamW |
| Learning rate | 5e-7 | 5e-7 | 1e-5 | 1e-5 | 1e-5 | 1e-5 | 4e-5 |
| $\beta$ | (0.9, 0.999) | (0.9, 0.999) | (0.9, 0.95) | (0.9, 0.95) | (0.9, 0.95) | (0.9, 0.95) | (0.9, 0.95) |
| weight decay | 0.01 | 0.01 | 0.02 | 0.02 | 0.05 | 0.05 | 0 |
| warmup steps | 1000 | 1000 | 500 | 500 | 50 | 50 | 1000 |
| Training steps | 5K | 5K | 3K | 5K | 5K | 5K | 1K |
| Grad. accumulation | 5 | 5 | 2 | 2 | 16 | 16 | 1 |
| Per-GPU batch size | 4 | 2 | 96 | 48 | 42 | 20 | 8 (6 GPUs) |
| Training time (hours) | 4 | 9 | 7 | 12.5 | 2 | 3 | 4.5 |
| **Trainable modules** | | | | | | | |
| Frozen parts | CLIP | | — | | MLLM | | Und. Expert |
| **Loss weights** | | | | | | | |
| $\lambda_{i2t}$ | 1.0 | | 1.0 | | 0.0 | | 0.0 |
| $\lambda_{\text{RECA}}$ | 1.0 | | 1.0 | | 1.0 | | 1.0 |

Forest, 2024), Playground-v3 (Liu et al., 2024a), and DALL-E 3 (Betker et al., 2023). (II) *Unified multimodal models*: Show-o (Xie et al., 2025b), Harmon (Wu et al., 2025d), Janus-pro (Chen et al., 2025d), OmniGen2 (Wu et al., 2025b), BLIP3-o (Chen et al., 2025b), BAGEL (Deng et al., 2025), Show-o2 (Xie et al., 2025c), UniWorld-V1 (Lin et al., 2025), Ovis-U1 (Wang et al., 2025a), and GPT-4o-Image (OpenAI, 2024). For enhanced statistical reliability, we reproduced some models using 12 random seeds (tripling the standard 4 seeds).

For image editing evaluation, we employ ImgEdit (Ye et al., 2025) and GEdit-Bench-EN (Liu et al., 2025) benchmarks. We exclude GPT-4o-Image and its distillation-trained models due to poor identity preservation. We select BAGEL (Deng et al., 2025), and FLUX-Kontext (Labs et al., 2025) as our primary baselines. Notably, we compare with the concurrent work BAGEL-NHR (Kuprashevich et al., 2025), which employs supervised fine-tuning (SFT) using 300,000 high-quality image editing data. Due to changes in the GPT API version and benchmark maintenance issues, the leaderboard scores exhibit significant variance and are difficult to reproduce. Therefore, we follow BAGEL (Deng et al., 2025) and report our own local evaluations for consistency. Following the original benchmark, we use gpt-4.1-2025-04-14 for ImgEdit and GEdit-Bench-EN evaluation.

**Training Data.** Due to limited availability of UMM training data, we use the high-quality open-source MidjourneyV6 dataset (CortexLM, 2024) (MIT License) for Show-o, Harmon, and OpenUni. For BAGEL, initial experiments using MidjourneyV6 data led to significantly higher initial losses and degraded image quality, which we attribute to distribution mismatch because MidjourneyV6 images are outside BAGEL's original training distribution. Therefore, we use 10,000 $1024 \times 1024$ FLUX-generated (BlackForest, 2024) images from Text-to-Image-2M (jackyhate, 2024) (MIT License), as BAGEL was originally trained on FLUX-generated images (Deng et al., 2025), ensuring better distribution alignment and training stability.

As we discussed in Sec. 2.2, to preserve visual understanding capabilities of UMMs with shared parameters for both understanding and generation (Show-o and Harmon), we also incorporate LLaVA Mix-665K (Liu et al., 2024b) (CC BY 4.0); for BAGEL and OpenUni, we set $\lambda_{i2t} = 0$.

GPT-4o-Image distillation data (e.g., BLIP3o-60k) can boost GenEval and DPGBench scores by 5~7% on average (Chen et al., 2025b). However, these datasets contain GenEval prompt templates, creating evaluation bias which we call **GenEval Template Leakage**. In our primary experiments, we deliberately avoid BLIP3o-60k to ensure fair comparison. We provide a detailed analysis of this leakage issue in Sec F.1.

## F   MORE EXPERIMENTAL RESULTS AND ANALYSIS

We present additional qualitative text-to-image generation and image editing results in Figure 8 and Figure 9. The detailed captions of T2I results are listed in Appendix H.

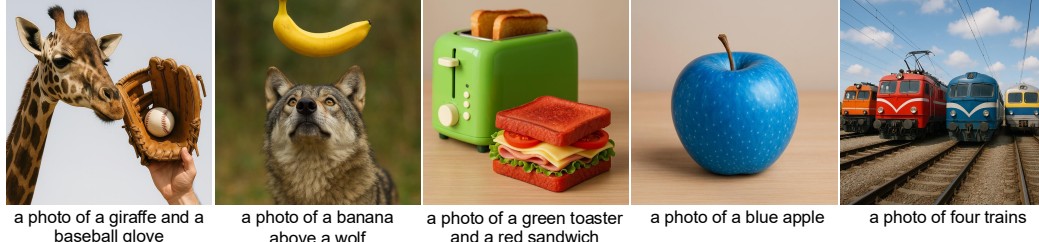

a photo of a giraffe and a baseball glove     a photo of a banana above a wolf     a photo of a green toaster and a red sandwich     a photo of a blue apple     a photo of four trains

Figure 13: **GenEval template leakage in BLIP3o-60k.** Examples from BLIP3o-60k that mirror GenEval templates, creating evaluation bias for models trained on this dataset.

## F.1 GENEVAL TEMPLATE LEAKAGE

GenEval (Ghosh et al., 2023) is a text-to-image evaluation benchmark which includes six sub-tasks: *Single Object*, *Two Objects*, *Color*, *Color Attributes*, *Counting*, and *Position*. The templates for these six sub-tasks are as follows:

- Single Object: `a photo of a {object}`.
- Two Objects: `a photo of a {object1} and a {object2}`.
- Color: `a photo of a {color} {object}`.
- Color Attri.: `a photo of a {color} {object1} and a {color} {object2}`.
- Counting: `a photo of {count} {object}`.
- Position: `a photo of a {object} {spatial relationship} {object}`.

However, we find that the BLIP3o-60k dataset (Chen et al., 2025b), distilled from GPT-4o-Image, contains 7,000 text–image pairs that nearly duplicate these templates. As shown in Figure 13, many training examples align closely with GenEval prompts, raising a serious risk of evaluation contamination. Models trained on BLIP3o-60k may gain unfair advantages by having effectively seen test-like data during training.

To ensure fair comparison, we evaluate both the full BLIP3o-60k dataset and a cleaned version with these 7,000 pairs removed. With the full set, SFT attains an inflated GenEval score of 84.95, but after removal drops to 80.88 (-4.07). In contrast, DPGBench remains unchanged (+0.05), confirming that these pairs only boost GenEval-specific performance rather than general generation capability. Thus, BLIP3o-60k reflects benchmark-targeted instruction tuning rather than a universal solution.

**RECA achieves comparable performance and superior robustness.** Using the full BLIP3o-60k dataset, RECA reaches 85.21 on GenEval, matching the benefit of SFT while requiring only unlabeled data and self-supervision. Crucially, on DPGBench RECA surpasses SFT (86.50 vs. 85.19), despite BLIP3o-60k containing dense-prompt data favorable to SFT. When 7,000 data are removed, RECA declines by only 0.45 and retains its DPGBench score. This robustness underscores that RECA provides a generalizable alignment signal, avoiding benchmark leakage and overfitting.

## F.2 RECA ON SHOW-O WITH IMAGE TOKENIZATION

**Naive $512 \times 512$ reconstruction collapses.** When we do RECA on the VQGAN variant of Show-o at $512 \times 512$ resolution, which uses a VQGAN (Esser et al., 2021) image tokenizer as the visual understanding encoder, the cross-entropy loss of RECA rapidly drops to nearly zero and the model's performance degrades, as shown in Figure 14. We observe that the model simply copies tokens from the input sequence instead of learning a meaningful conditioning. To mitigate this token-copying behavior, we explore three strategies:

- Resizing input images to $256 \times 256$, which aligns with the configuration in Sec. 2.2 and feeds the visual encoder its minimum supported resolution.
- Resizing input images to $128 \times 128$.

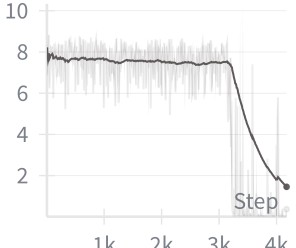

| Setting | GenEval ↑ | DPG ↑ | WISE ↑ |
|---|---|---|---|
| Baseline (no RECA) | 68.70 | 83.63 | 0.40 |
| 512×512 | 66.59 | 83.55 | 0.30 |
| 256×256 | 70.61 | 84.77 | **0.40** |
| 128×128 | 67.47 | 83.22 | 0.39 |
| Blur | **72.20** | **85.12** | 0.38 |

Figure 14: **Token-copying collapse and mitigation for Show-o's VQGAN-understanding variant.** *Left:* Cross-entropy loss plunges after 3,000 steps when training at $512 \times 512$. *Right:* Quantitative results of Show-o under different reconstruction inputs.

- Blurring input images by downsampling it by $8\times$ and then upsampling back.

The blurred reconstruction method yields the highest GenEval and DPGBench scores but slightly deteriorates WISE reasoning accuracy. Resizing inputs to $256\times256$ offers the best balance across all metrics. In contrast, $128 \times 128$ inputs fall outside Show-o's pretraining distribution and significantly reduce all of these scores.

**Why the gain of VQGAN variant is smaller than that of CLIP variant?** Notably, the tokenizer variant starts from higher performance (68.7/83.63 *vs.* 66.2/82.21), yet after RECA the 256×256 reconstruction reaches only 70.61/84.77 compared with 72.3/84.94 for the Show-o CLIP variant. This gap is consistent with the weaker semantic representations provided by the VQGAN tokenizer and the lower visual understanding capacity of the Show-o tokenizer variant, leaving RECA with less high-level signal to leverage.

### F.3 COMPREHENSIVE QUANTITATIVE RESULTS

We provide comprehensive quantitative analysis of RECA across all evaluated unified multimodal models in this section. Table 9 shows the full comparison between RECA post-trained Harmon-1.5B and previous models. Table 10 shows RECA's consistent performance improvements across all evaluated architectures, with particularly notable gains in positional understanding and color attribution tasks. The detailed WISE benchmark results in Table 11 reveal that RECA primarily enhances semantic alignment capabilities while showing modest improvements in reasoning-intensive tasks.

### F.4 ADDITIONAL COMPOSITIONAL EVALUATION ON T2I-COMPBENCH

To further substantiate the robustness of our evaluation framework and address concerns regarding the complexity of the compositional tests, we conducted an additional rigorous analysis utilizing the **T2I-CompBench** (Huang et al., 2023). T2I-CompBench is a specialized dataset comprising 2,670 carefully curated test samples designed to systematically evaluate the compositional understanding capabilities of text-to-image models across **10 distinct compositional dimensions**. While GenEval primarily focuses on four fundamental attributes (object presence, count, color, and 2D position), T2I-CompBench systematically assesses a comprehensive range of high-level compositional attributes and relationships. Specifically, T2I-CompBench includes critical capability dimensions absent or underspecified in simpler benchmarks, such as:

- **Shape & Texture:** Dedicated tests for understanding diverse geometric forms (e.g., "a pentagonal stop sign") and material properties (e.g., "a plastic toy").

- **Numeracy:** Precise control over object quantities (e.g., "six airplanes").

- **3D Spatial:** Crucially, the benchmark distinguishes between 2D spatial and 3D spatial reasoning, with the latter focusing specifically on depth perception and occlusion relationships (e.g., "a chair hidden by a turtle")—a key capability for evaluating complex scene understanding.

- **Complex Scenes:** High-level tasks like complex scenes and complex action/spatial scenarios evaluate models' generalization capability in situations involving multiple intertwined attributes,

Table 9: **Results on GenEval and DPGBench**. Scores marked with (*) are our reproduced results using 12 random seeds. We use Harmon-1.5B as our base model and post-train it with RECA. The gray-colored rows denote private models, and their results are cited from (Yan et al., 2025b; Geng et al., 2025). Arrows (↑) denote that higher is better.

| Model | Params | GenEval ↑ | | | | | | | DPG↑ |
| --- | --- | --- | --- | --- | --- | --- | --- | --- | --- |
| | | Single Obj. | Two Obj. | Counting | Colors | Position | Color Attri. | Overall | Score |
| *Generation Only Models* | | | | | | | | | |
| SD3-Medium | 2B | 0.99 | 0.94 | 0.72 | 0.89 | 0.33 | 0.60 | 0.74 | 84.08 |
| SDXL | 2.6B | 0.98 | 0.74 | 0.39 | 0.85 | 0.15 | 0.23 | 0.55 | 74.65 |
| SANA-1.5 | 4.8B | 0.99 | 0.93 | 0.86 | 0.84 | 0.59 | 0.65 | 0.81 | 84.70 |
| Emu3-Gen | 8B | 0.98 | 0.71 | 0.34 | 0.81 | 0.17 | 0.21 | 0.54 | 80.60 |
| FLUX-dev | 12B | 0.99 | 0.85 | 0.74 | 0.79 | 0.21 | 0.48 | 0.68 | 84.00 |
| Playground-v3 | 24B | 0.99 | 0.95 | 0.72 | 0.82 | 0.50 | 0.54 | 0.76 | 87.06 |
| DALL-E 3 | – | 0.96 | 0.87 | 0.47 | 0.83 | 0.43 | 0.45 | 0.67 | 83.50 |
| *Unified Multimodal Models* | | | | | | | | | |
| GPT-4o-Image | - | 0.99 | 0.92 | **0.85** | 0.89 | 0.74 | 0.71 | 0.84 | 86.23 |
| Show-o* | 1.3B | 0.98 | 0.84 | 0.67 | 0.82 | 0.30 | 0.52 | 0.69 | 83.63 |
| Harmon* | 1.5B | 0.99 | 0.87 | 0.69 | 0.86 | 0.45 | 0.51 | 0.73 | 80.93 |
| Show-o2 | 7B | 1.00 | 0.87 | 0.58 | 0.92 | 0.52 | 0.62 | 0.76 | 86.14 |
| Janus-Pro | 7B | 0.99 | 0.89 | 0.59 | 0.90 | **0.79** | 0.66 | 0.80 | 84.33 |
| BAGEL* | 14B | 0.99 | 0.93 | 0.80 | 0.86 | 0.51 | 0.63 | 0.79 | 84.03 |
| **RECA** | 1.5B | 1.00 | **0.98** | 0.71 | **0.93** | 0.76 | **0.77** | **0.86** | **87.21** |
| *Unified Multimodal Models Trained with GPT-4o Data* | | | | | | | | | |
| Ovis-U1 | 3.6B | 0.98 | **0.98** | **0.90** | 0.92 | 0.79 | 0.75 | 0.89 | 83.72 |
| OmniGen2 | 7B | 1.00 | 0.95 | 0.64 | 0.88 | 0.55 | 0.76 | 0.80 | 83.57 |
| BLIP3-o* | 8B | 1.00 | 0.92 | 0.63 | 0.91 | 0.86 | 0.67 | 0.83 | 80.73 |
| UniWorld-V1 | 20B | 0.99 | 0.93 | 0.79 | 0.89 | 0.49 | 0.70 | 0.80 | – |
| **RECA** | 1.5B | 1.00 | 0.97 | 0.76 | **0.94** | **0.91** | **0.83** | **0.90** | **88.15** |

Table 10: **RECA brings consistent performance gains to various UMM frameworks.** We show the performance gains after applying RECA to different unified multimodal models. All models are evaluated on GenEval, DPGBench, and WISE benchmarks.

| Model | RECA | GenEval | | | | | | | DPG |
| --- | --- | --- | --- | --- | --- | --- | --- | --- | --- |
| | | Single | Two | Count | Color | Position | Attri. | Overall | |
| Show-o-256 | ✗ | **97.4** | 63.3 | 52.1 | 82.3 | 14.2 | 30.3 | 56.6 | 70.65 |
| | ✓ | **97.4** (0.0) | 73.6 (+10.3) | 56.0 (+3.9) | 83.8 (+1.5) | 20.3 (+6.1) | 40.2 (+9.9) | 61.9 (+5.3) | 75.70 (+5.05) |
| Show-o-512 | ✗ | 97.2 | 80.3 | 61.9 | 78.2 | 27.3 | 52.3 | 66.2 | 82.21 |
| | ✓ | 98.2 (+1.0) | 90.6 (+10.3) | 66.8 (+4.9) | 84.1 (+5.9) | 37.4 (+10.1) | 56.8 (+4.5) | 72.3 (+6.1) | 84.94 (+2.73) |
| OpenUni-1.6B | ✗ | 96.8 | 63.3 | 46.4 | 80.1 | 18.5 | 30.8 | 56.0 | 76.29 |
| | ✓ | 97.1 (+0.3) | 84.3 (+21.0) | 57.4 (+11.0) | 83.5 (+3.4) | 44.3 (+25.8) | 56.0 (+25.2) | 70.4 (+14.4) | 80.45 (+4.16) |
| OpenUni-3.6B | ✗ | **99.1** | 71.8 | 51.9 | 83.9 | 23.3 | 41.6 | 61.9 | 79.02 |
| | ✓ | **99.1** (0.0) | 92.7 (+20.9) | 52.3 (+0.4) | 87.1 (+3.2) | 43.8 (+20.5) | 70.3 (+28.7) | 74.1 (+12.2) | 82.75 (+3.73) |
| Harmon-0.5B | ✗ | 99.7 | 80.5 | 55.8 | 86.7 | 32.2 | 49.7 | 67.6 | 80.12 |
| | ✓ | 99.9 (+0.2) | 92.3 (+11.8) | 59.4 (+3.6) | 91.7 (+5.0) | 58.5 (+26.3) | 70.7 (+21.0) | 78.7 (+11.1) | 84.67 (+4.55) |
| Harmon-1.5B | ✗ | 99.4 | 87.3 | 68.7 | 86.4 | 44.9 | 51.1 | 72.9 | 80.93 |
| | ✓ | 99.9 (+0.5) | 97.7 (+10.4) | 71.4 (+2.7) | 92.6 (+6.2) | 75.7 (+30.8) | 76.6 (+25.5) | 85.7 (+12.8) | 87.21 (+6.28) |
| BAGEL | ✗ | 99.1 | 93.0 | 79.9 | 86.0 | 51.3 | 63.4 | 78.8 | 84.03 |
| | ✓ | 99.3 (+0.2) | 93.9 (+0.9) | 80.3 (+0.4) | 87.6 (+1.6) | 60.8 (+9.5) | 72.6 (+9.2) | 82.4 (+3.6) | 85.29 (+1.26) |

objects, and abstract relationships (e.g., "The red hat was on top of the brown coat rack" simultaneously testing color binding, spatial relationships, and object recognition).

Following SRUM (Jin et al., 2025), we employ QwenVL-2.5-32B (Bai et al., 2025) as the designated multimodal evaluator. The results presented in Table 12 validate the effectiveness of RECA across all tested UMM architectures, consistently delivering substantial performance gains. Crucially, the results on T2I-CompBench serve as a powerful complement to our GenEval findings. While GenEval confirms alignment in fundamental attributes (Color) and 2D positioning, T2I-CompBench demonstrates that our method's improvements generalize to a much broader spectrum of visual semantics. **(I) Beyond Color:** The substantial gains in **Texture** and **Shape** (e.g., +10.38 on shape for OpenUni) prove that RECA refines the model's sensitivity to all fine-grained visual attributes, not just

Table 11: **WISE benchmark results**. Performance on reasoning-based text-to-image generation. Harmon and OpenUni exhibit zero-shot improvement but Show-o and BAGEL do not.

| Model | RECA | Cultural | Time | Space | Biology | Physics | Chemistry | Overall |
|---|---|---|---|---|---|---|---|---|
| Show-o-256 | ✗ | 0.27 | 0.38 | 0.45 | 0.26 | 0.37 | 0.24 | 0.33 |
| | ✓ | **0.28** (+0.01) | **0.40** (+0.02) | **0.46** (+0.01) | **0.27** (+0.01) | **0.38** (+0.01) | 0.24 (0.00) | **0.34** (+0.01) |
| Show-o-512 | ✗ | 0.31 | 0.42 | 0.55 | 0.31 | 0.48 | **0.32** | 0.40 |
| | ✓ | 0.31 (0.00) | 0.42 (0.00) | **0.56** (+0.01) | **0.32** (+0.01) | **0.50** (+0.02) | 0.30 (-0.02) | 0.40 (0.00) |
| OpenUni-1.6B | ✗ | 0.40 | 0.41 | 0.54 | 0.35 | 0.52 | 0.36 | 0.43 |
| | ✓ | 0.40 (0.00) | **0.46** (+0.05) | **0.56** (+0.02) | **0.42** (+0.07) | **0.54** (+0.02) | 0.34 (-0.02) | **0.45** (+0.02) |
| OpenUni-3.6B | ✗ | 0.36 | 0.38 | 0.59 | 0.43 | 0.50 | 0.34 | 0.43 |
| | ✓ | **0.52** (+0.16) | **0.51** (+0.13) | **0.69** (+0.10) | **0.50** (+0.07) | **0.64** (+0.14) | **0.40** (+0.06) | **0.54** (+0.11) |
| Harmon-0.5B | ✗ | 0.28 | 0.39 | 0.41 | 0.31 | 0.38 | 0.22 | 0.33 |
| | ✓ | **0.33** (+0.05) | **0.43** (+0.04) | **0.54** (+0.13) | **0.38** (+0.07) | **0.48** (+0.10) | **0.24** (+0.02) | **0.40** (+0.07) |
| Harmon-1.5B | ✗ | 0.38 | 0.48 | 0.52 | 0.37 | 0.44 | 0.29 | 0.41 |
| | ✓ | **0.44** (+0.06) | **0.58** (+0.10) | **0.57** (+0.05) | **0.48** (+0.11) | **0.58** (+0.14) | **0.32** (+0.03) | **0.50** (+0.09) |
| BAGEL | ✗ | 0.42 | **0.53** | 0.64 | 0.42 | 0.57 | 0.43 | 0.50 |
| | ✓ | **0.43** (+0.01) | 0.51 (-0.02) | **0.67** (+0.03) | **0.46** (+0.04) | **0.59** (+0.02) | **0.46** (+0.03) | **0.52** (+0.02) |

Table 12: **T2I-CompBench Performance Analysis.** Comparison of performance across subcategories before and after applying RECA.

| Category | OpenUni-3.6B | | Harmon-1.5B | | Show-o-512 | | BAGEL | |
|---|---|---|---|---|---|---|---|---|
| | Base | RECA | Base | RECA | Base | RECA | Base | RECA |
| Color | 87.47 | **92.50** (+5.03) | 89.55 | **93.72** (+4.17) | 88.60 | **90.88** (+2.28) | 93.05 | **93.62** (+0.57) |
| Shape | 75.22 | **85.60** (+10.38) | 79.09 | **84.32** (+5.23) | 77.68 | **82.22** (+4.54) | 84.25 | **85.72** (+1.47) |
| Texture | 82.87 | **89.65** (+6.78) | 86.28 | **91.37** (+5.09) | 85.23 | **88.37** (+3.14) | 88.95 | **90.58** (+1.63) |
| Spatial | 78.29 | **86.30** (+8.01) | 84.98 | **92.37** (+7.39) | 85.75 | **87.28** (+1.53) | 88.42 | **88.57** (+0.15) |
| Non-Spatial | 83.10 | **87.18** (+4.08) | 82.77 | **87.12** (+4.35) | 85.38 | 85.13 (-0.25) | 88.27 | **88.80** (+0.53) |
| Numeracy | 64.44 | **75.97** (+11.53) | 69.19 | **79.78** (+10.59) | 71.39 | **74.42** (+3.03) | 79.43 | **81.88** (+2.45) |
| 3D Spatial | 68.60 | **84.35** (+15.75) | 75.58 | **86.08** (+10.50) | 75.38 | **81.42** (+6.04) | 81.47 | **83.50** (+2.03) |
| Complex | 82.82 | **87.79** (+4.97) | 82.05 | **88.51** (+6.46) | 82.73 | **85.07** (+2.34) | 87.85 | **88.74** (+0.89) |
| **Overall** | 78.84 | **86.49** (+7.65) | 81.36 | **88.03** (+6.67) | 82.12 | **84.47** (+2.35) | 86.74 | **87.89** (+1.15) |

color binding. **(II) Beyond 2D Layout:** The remarkable improvements in **3D Spatial** reasoning (e.g., +15.75 for OpenUni, +10.50 for Harmon) indicate that RECA effectively teaches the model to comprehend depth, perspective, and occlusion, far exceeding simple up/down/left/right positional alignment. **(III) In Complex Scenario**, all models showed consistent gains (e.g., Harmon-1.5B increased by +6.46 points). The only minor fluctuation was observed in the Non-Spatial category for Show-o-512 (−0.25), which does not diminish the overwhelming positive trend across high-difficulty tasks.

## F.5 GENEVAL SUBTASK DYNAMICS

**How do individual GenEval subtasks evolve?** Figure 15 shows that RECA primarily boosts performance within the first $3k \sim 5k$ steps before convergence. *Single Object* and *Two Objects* success rates are already saturated and quickly flatten once they touch the ceiling. *Color* continues to climb through the entire run, indicating sustained improvements in attribute grounding. *Position* rises sharply before 5k steps and then stabilizes at its ceiling. *Color Attributes* accuracy peaks around 5k steps and then gently drops as we optimize for other objectives. *Counting* exhibits only minor fluctuations without a clear upward trend, matching the limitation we discuss in Appendix G. Understanding how to balance different subtasks for finer-grained control is an important direction for future work.

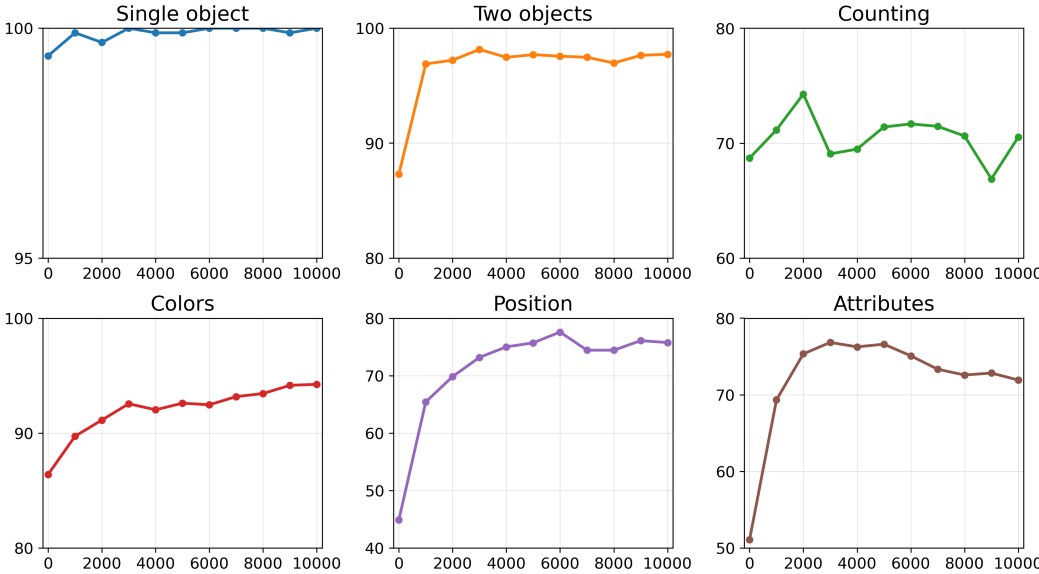

Figure 15: **GenEval subtask performance dynamic during RECA training.** We plot the Harmon-1.5B scores for each GenEval component at different training steps.

### F.6 FURTHER DISCUSSION ON EXPERIMENT RESULTS

**Why the improvements on Show-o is modest?** (I) The CLIP used in Show-o exhibits insufficient high-level semantic information (Tong et al., 2024b), which limits the upperbound of RECA. (II) Show-o's codebook has only 4096 tokens. fundamentally limits its representational capacity and generation performance.

**Why the text-to-image generation improvements on BAGEL is modest?** (I) The data and computational resources prevent the full exploitation of the potential benefits of RECA. (II) BAGEL's pre-existing image editing training, which conditions on both SigLIP (Zhai et al., 2023) and VAE (Kingma & Welling, 2013) features, has already endowed the model with good capabilities for semantic-level reconstruction. It constrains the improvement space in generation tasks, while correspondingly enabling substantial enhancements in image editing capabilities.

### F.7 MORE VISUALIZATION OF RECONSTRUCTION RESULTS

To provide deeper insights into how RECA improves model capabilities, we visualize the reconstruction results. Figure 16 demonstrates how Harmon's reconstruction quality evolves across different training timesteps, while Figure 17 compares reconstruction quality before and after RECA post-training across all evaluated UMM architectures.

As shown in Figure 17, RECA post-training yields substantial improvements in reconstruction quality across all evaluated architectures:

- **Show-o**: Reconstruction transitions from almost noisy outputs to clearer images with better preservation of semantic content and object structures.
- **OpenUni, Harmon**: Reconstruction quality improves significantly, with enhanced detail preservation and more accurate rendering of geometry, color and texture.
- **BAGEL**: Despite already having some reconstruction capability from pre-training, RECA further enhances reconstruction quality, particularly in preserving fine-grained semantic details and spatial relationships.

**Training progression analysis.** Figure 16 shows how Harmon's reconstruction ability improves over the course of RECA training. Early outputs are blurry and lack clear structure, but as training progresses, the model learns to recover object shapes, colors, and layouts more accurately. By the end, the reconstructions are sharp and semantically faithful.

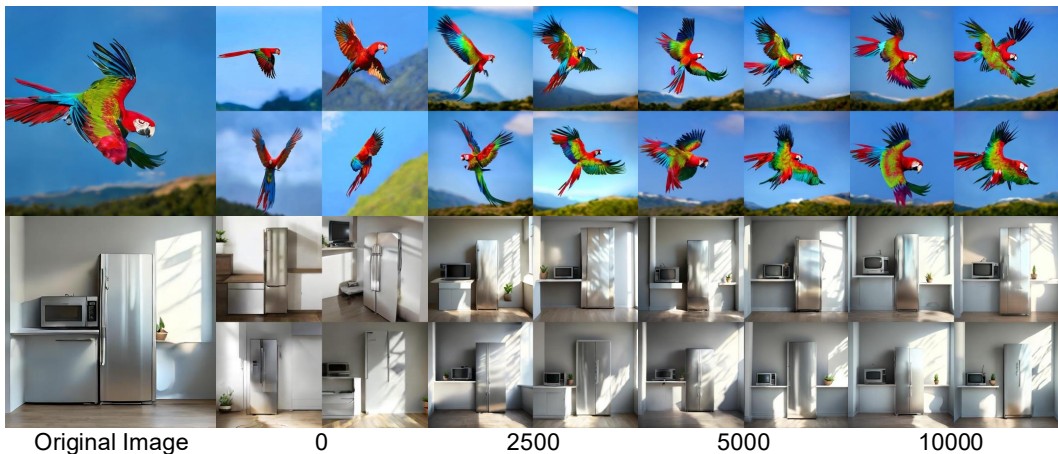

Figure 16: **Harmon reconstruction progression during RECA training.** (Left to Right) Early timesteps show blurry or incomplete reconstructions, while later stages achieve high-fidelity reconstruction with accurate semantic details, object structures, and color rendering.

## F.8 MORE ABLATION STUDIES

To evaluate the robustness of RECA, we conduct comprehensive ablation studies on the scale and source of the training data.

**Data Scale Scalability.** We first evaluate Harmon-1.5B with varying amounts of MidjourneyV6 (CortexLM, 2024) data, ranging from 10,000 to 240,000. As shown in Table 13, RECA demonstrates consistent performance gains as the data scale increases, highlighting its potential for further improvement with larger datasets.

**Data Source Robustness.** Crucially, we investigate whether RECA relies on high-quality synthetic data. We select three distinct datasets representing different domains and quality levels: **(I) COCO2014** (Lin et al., 2014), representing *real-world* data with relatively *lower* visual quality; **(II) JourneyDB** (Sun et al., 2023), representing synthetic data with *medium* quality; **(III) BLIP3o-60k** (Chen et al., 2025a) and **(IV) MidjourneyV6** (CortexLM, 2024), representing *high-quality* synthetic data. As shown in Table 14, RECA maintains stable and robust performance across all sources. Notably, even when trained on lower-quality real-world images (COCO), the method yields competitive improvements (GenEval 85.3), indicating that RECA is applicable to broad data distributions.

Table 13: **Data scale analysis.** Performance with varying amounts of MidjourneyV6 data.

| Data | Scale | GenEval | DPG |
|------|-------|---------|-----|
| MidJourneyv6 | 10,000 | 84.6 | 86.82 |
| MidJourneyv6 | 50,000 | 85.3 | 87.06 |
| MidJourneyv6 | 240,000 | 85.7 | 87.21 |

Table 14: **Data source robustness.** Performance across different data qualities.

| Data | Quality | GenEval | DPG |
|------|---------|---------|-----|
| COCO2014 | Low | 85.3 | 86.82 |
| JourneyDB | Medium | 85.2 | 86.71 |
| BLIP3o | High | 85.2 | 86.50 |
| MidJourneyv6 | High | 85.7 | 87.21 |

## G LIMITATIONS AND FUTURE WORK

Despite the effectiveness of RECA across various unified multimodal models (UMMs), our approach has several limitations that warrant discussion:

**The improvements on the counting task are minor.** As shown in Table 10, the model's gains on counting are not substantial. We attribute this to object number being a mid-level visual feature that current MLLMs are not adept at extracting. BLINK (Fu et al., 2024) points out that the information required by high-level semantic tasks (e.g., distinguishing left from right, identifying color) can often be captured and leveraged in linguistic form, whereas counting resists language-only mediation in

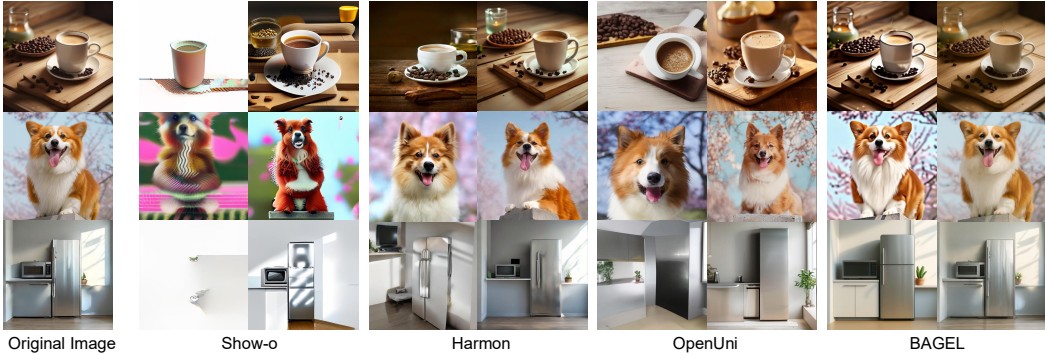

Figure 17: **Reconstruction comparison across UMM architectures.** For each model (Show-o, Harmon, OpenUni, BAGEL), we show reconstruction results before (left) and after (right) RECA post-training. RECA consistently improves reconstruction fidelity, especially in semantic details and color accuracy, which further benefits text-to-image generation.

MLLMs. Consequently, RECA primarily enhances the model's generative ability on tasks directly tied to high-level semantics, such as color attributes and positional relations. Improving counting may require the mixture training objective of high-quality counting datasets or the incorporation of reinforcement learning techniques.

**Architecture-specific constraints.** RECA still exhibits limited effectiveness on BLIP-3o. As illustrated in Figure 18, BLIP-3o possesses inherent image reconstruction capabilities. Application of RECA fails to improve reconstruction or text-to-image generation quality, likely because BLIP-3o's pre-training already incorporates strong reconstruction objectives, rendering additional reconstruction training counterproductive.

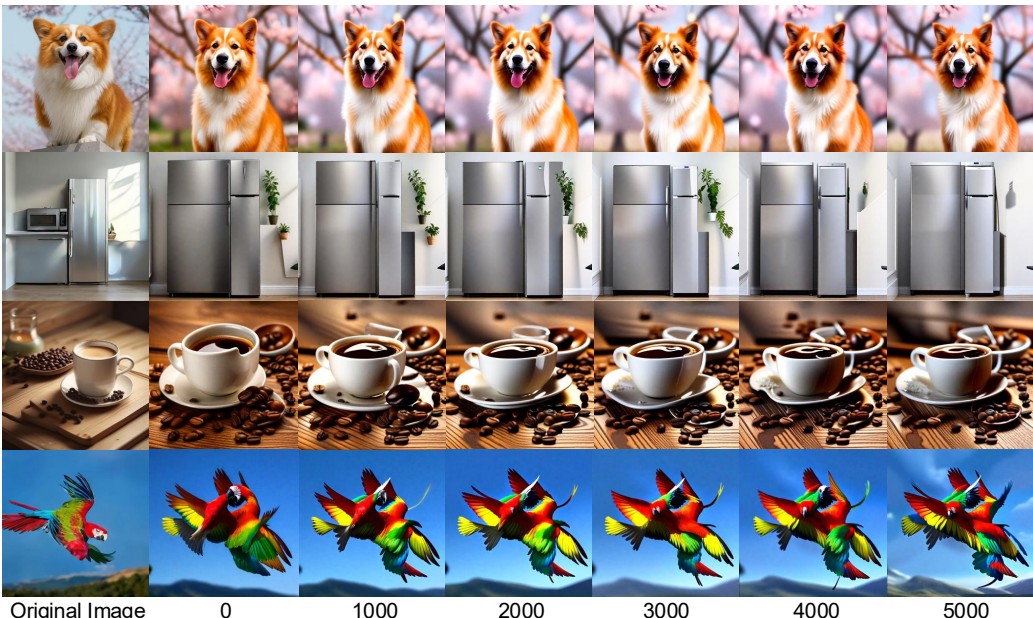

Figure 18: **BLIP-3o reconstruction results.** We observe that BLIP-3o possesses inherent image reconstruction capabilities. Following RECA application, reconstruction quality exhibits degradation rather than improvement, suggesting that BLIP-3o incorporates reconstruction objectives during pre-training.

## H  GENERATED CAPTIONS

### H.1  CAPTIONS FOR FIGURE 2

**10-word Caption**: Kitten and puppy cuddle warmly under a cozy yellow blanket.

**100-word Caption**: A heartwarming scene of pure companionship unfolds on a soft white bed where a tiny tabby kitten and a tricolor Corgi puppy nestle side by side. Wrapped gently under a fluffy mustard-yellow blanket, they gaze curiously at the viewer with wide, innocent eyes. The kitten's delicate stripes and the puppy's soft fur blend harmoniously under the golden tones of the blanket, creating a tender atmosphere filled with comfort and warmth. Their small heads rest close, touching lightly, as if they've found solace in each other's presence. The peaceful bedroom backdrop adds a serene touch to this moment of affection.

**500-word Caption**: In a warmly lit bedroom, the tranquil quiet of the morning is broken only by the soft breaths of two unlikely yet inseparable companions. Upon a pristine white bed, layered with thick cotton sheets and plush pillows, lies a tender tableau of comfort and innocence: a tabby kitten and a tricolor Corgi puppy, curled gently together beneath a soft, fleece mustard-yellow blanket. Their eyes are wide and bright, reflecting a mixture of curiosity and calm, as if they've just woken up from a shared dream. The kitten is small, delicate, with a soft grayish-brown tabby pattern running across its fur like brushstrokes. Its tiny paws are tucked beneath its chest, body pressed closely to the puppy's for warmth and security. Its ears point upward slightly, twitching at the soft sounds of the morning breeze filtering through the room. Its eyes, a bright glimmering hazel, look out with an expression that is both alert and serene. The puppy, a young Pembroke Welsh Corgi, has a face full of character—expressive brown eyes, upright ears edged with tan, and a snout speckled with white. Its fur is a beautiful tricolor blend of rich black, creamy tan, and snowy white, each hue carefully painted by nature to highlight its playful features. Though still small, its body is slightly larger than the kitten's, and its front paw is placed protectively near its feline friend. There is no tension between them—only the shared comfort of closeness, the unspoken language of trust. The blanket they share adds a golden warmth to the scene, its texture soft and inviting. It wraps around them like a cocoon, hinting at a chill in the air beyond the bed, but ensuring they remain snug in their haven. The folds of the blanket frame them like the petals of a flower, drawing the eye inward to their peaceful gaze and gentle postures. In the background, the bedroom is softly lit by morning light diffused through a window. The white and cream tones of the bedding and walls contribute to the serenity of the scene, allowing the vivid colors of the animals and their blanket to take center stage. Pillows are propped up neatly behind them, hinting that someone may have just stepped away, leaving behind these two nestled in a perfect moment of quiet affection. This image tells a silent story—not just of two animals coexisting, but of a genuine bond. It evokes themes of comfort, friendship, and the uncomplicated joy of presence. It reminds us of the purity in companionship, of how safety and peace can be found not just in familiar places, but in the closeness of those we trust. The kitten and the puppy, young and full of life, are captured here in a fleeting, gentle stillness that speaks to the heart.

### H.2  CAPTIONS FOR FIGURE 8

$512 \times 512$ images (left to right, top to bottom):

- Two spiraling strands of rich, crimson-colored pasta rest elegantly on the surface of a polished dark wooden table, the grain of the wood accentuating their vibrant hue. This rustic Italian kitchen is bathed in the warm, golden light of the late afternoon sun, which highlights the intricate texture of the pasta. The table, set amidst traditional décor and terracotta pots filled with fresh herbs, offers a tranquil setting for this simple yet captivating culinary display.
- A golden squirrel and a bright magenta elephant in bright sunlight.
- During the warm glow of a dwindling summer evening, a particular fussy feline with distinctive calico markings is perched atop a garden table. The cat, seemingly indifferent to its surroundings, sports a pair of large, reflective aviator sunglasses that sit comically upon its small, furry face. Around the cat, there are scattered pots of blooming flowers, contributing to the charm of the scene, and in the background, hints of orange and pink skies are visible through the foliage.
- a highly intricate and vibrant cityscape that reflects a fusion of Moebius's imaginative design and Makoto Shinkai's detailed animation style. The streets are aglow with neon signs in a kaleidoscope

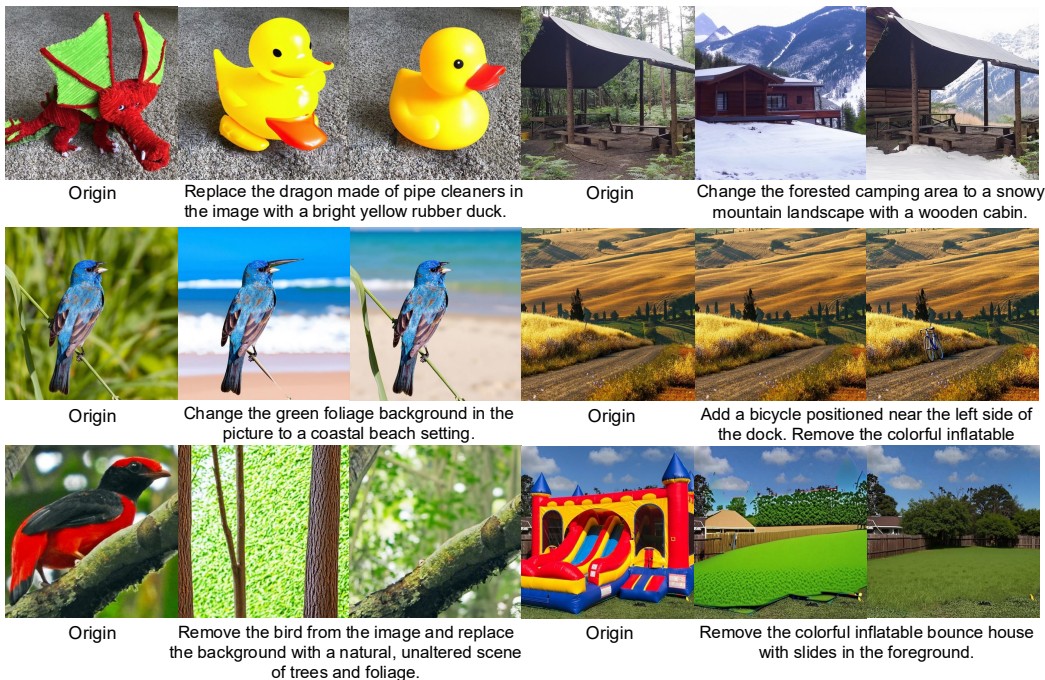

Figure 19: **Additional ImgEdit benchmark results.** Qualitative results on image editing tasks. In each pair, the left image is generated by the baseline model and the right by the RECA post-trained model

of colors, casting reflections on the glossy, rain-slicked pavements. Towering skyscrapers with glowing windows rise towards a starless night sky, as the artwork garners significant attention and praise on ArtStation.

- A deep red rose with plush petals sits elegantly coiled atop an ivory, intricately patterned lace napkin. The napkin rests on a rustic wooden table that contributes to the charming garden setting. As the late evening sun casts a warm golden hue over the area, the shadows of surrounding foliage dance gently around the rose, enhancing the romantic ambiance. Nearby, the green leaves of the garden plants provide a fresh and verdant backdrop to the scene.
- A close-up image capturing the intricate details of a maple leaf, which is composed entirely of clear, sparkling water droplets. The leaf is set against a smooth, dark background that accentuates its delicate water structure. The droplets glisten as they cling to the invisible veins of the leaf, creating a natural yet surreal piece of art.
- A detailed photograph captures the image of a statue with the likeness of an ancient pharaoh, unexpectedly accessorized with a pair of bronze steampunk goggles resting atop its head. The statue is dressed in an anachronistic fashion, featuring a crisp white t-shirt and a fitted black leather jacket that contrasts with its traditional headdress. The background is a simple, solid color that accentuates the statue's unconventional attire and the intricate details of the steampunk eyewear.
- On the soft, warm sand of the beach, a fluffy white rabbit with rounded ears is caught in a curious moment, gently placing its paw on the ribbed surface of a pink scallop shell. The scallop, slightly open, reveals its smooth interior contrasting with its coarse outer texture, while hues of pink and orange from the setting sun reflect off its surface. There's a tranquil ocean backdrop with the gentle ebbing of the tide, and the fading daylight casts a golden glow over the scene, highlighting the rabbit's soft fur and the shell's subtle color.

$1024 \times 1024$ images (left to right, top to bottom):

- A transparent glass cube on an endless desert, with a burning candle inside casting shadows on the sand.
- A drop of water containing a miniature forest with colorful tiny flower inside.

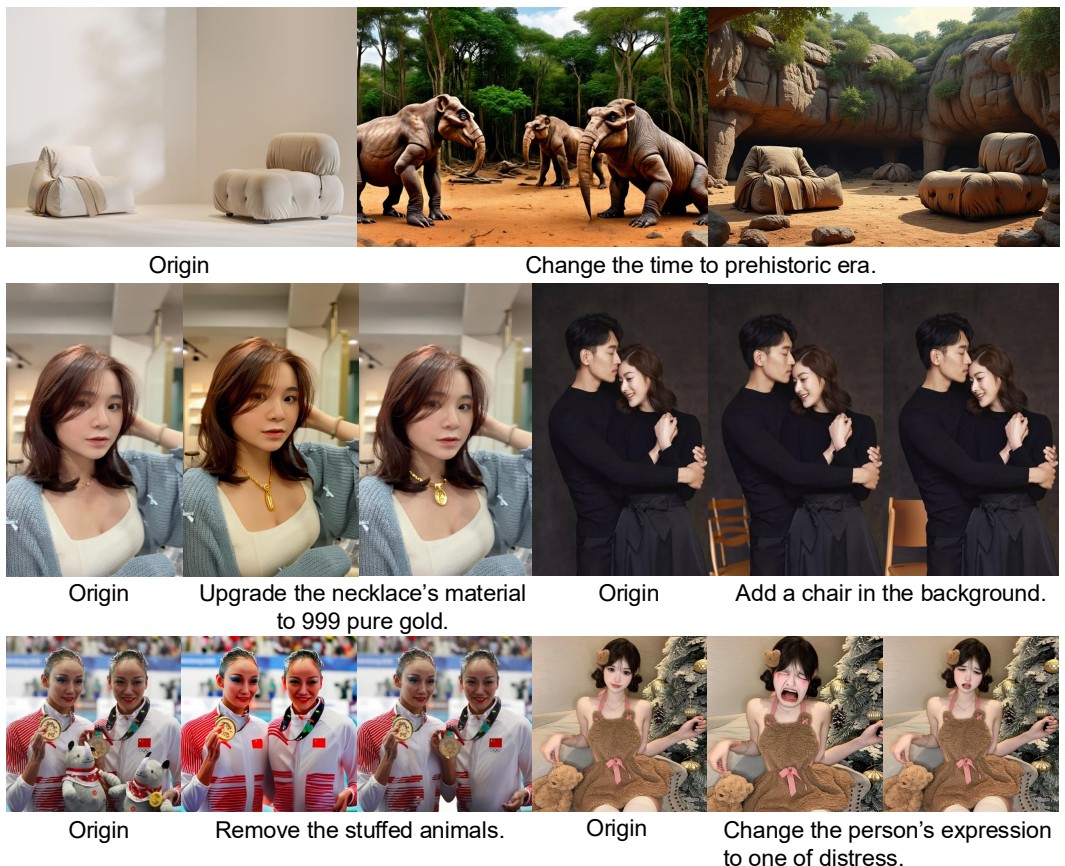

Figure 20: **Additional GEdit-Bench-EN results.** Qualitative results on image editing tasks. In each pair, the left image is generated by the baseline BAGEL model and the right by the RECA post-trained model.

- A vibrant and traditional depiction of Tteokguk, a Korean rice cake soup, served during the Chuseok festival. The image features a steaming bowl of clear broth filled with soft, chewy rice cakes, slices of zucchini, carrots, and green onions, garnished with a sprinkle of sesame seeds. Surrounding the bowl are traditional Korean elements, such as a woven basket with red dates, a small wooden spoon, and a wooden table with a warm, earthy tone. The atmosphere is cozy and festive, with soft, natural lighting and a slightly blurred background to emphasize the dish. The scene captures the essence of Korean culture and the warmth of the Chuseok celebration, with a focus on authenticity and detail.
- Photorealistic closeup image of two pirate ships battling each other as they sail inside a cup of coffee.
- A cute handmade felt doll of a little girl standing on a grassy patch. She wears an orange knitted hooded dress with blue buttons and matching boots. The girl is holding a smiling sun-shaped balloon made of felt. Fluffy white clouds with button details float in the sky, and soft green felt trees surround the scene. The style is whimsical, cozy, and playful, with pastel colors and a dreamy, handcrafted aesthetic.
- The word 'RECA' is written on a street surface, with the word 'STARTS' written just below it, surrounded by colorful chalk drawings and playful doodles.
- A Van Gogh style painting of a cyberpunk city at sunrise.
- An anime-style portrait of a girl with big sparkling eyes, detailed hair highlights, soft gradient background, vibrant colors.
- A dreamy composition of a young woman with butterflies emerging from her skin, wings glowing in soft golden hues, surreal and enchanting. She is wearing a wedding dress and has white angel wings, waving her hand.

- A surreal split-face portrait: left side realistic woman with soft skin and a vivid blue eye, right side robotic cyborg with exposed steel plates, fluorescent blue circuits, tiny gears, and a blood-red mechanical eye, cinematic lighting, futuristic and striking.

# I    MORE QUALITATIVE RESULTS ON IMAGE EDITING

We provide additional qualitative comparisons in this section. Figure 21 presents a comparison between the RECA post-trained BAGEL and previous SOTA models, ICEdit (Zhang et al., 2025c), FLUX-Kontext (Labs et al., 2025), and GPT-4o-Image. Compared with ICEdit and FLUX-Kontext, our method demonstrates stronger instruction-following capability. In contrast to GPT-4o-Image, our method exhibits superior identity preservation and background fidelity. Figure 19 and 20 showcase consistent improvements in semantic consistency, instruction following, and visual quality preservation across various editing tasks including background modification, style transfer, and object manipulation.

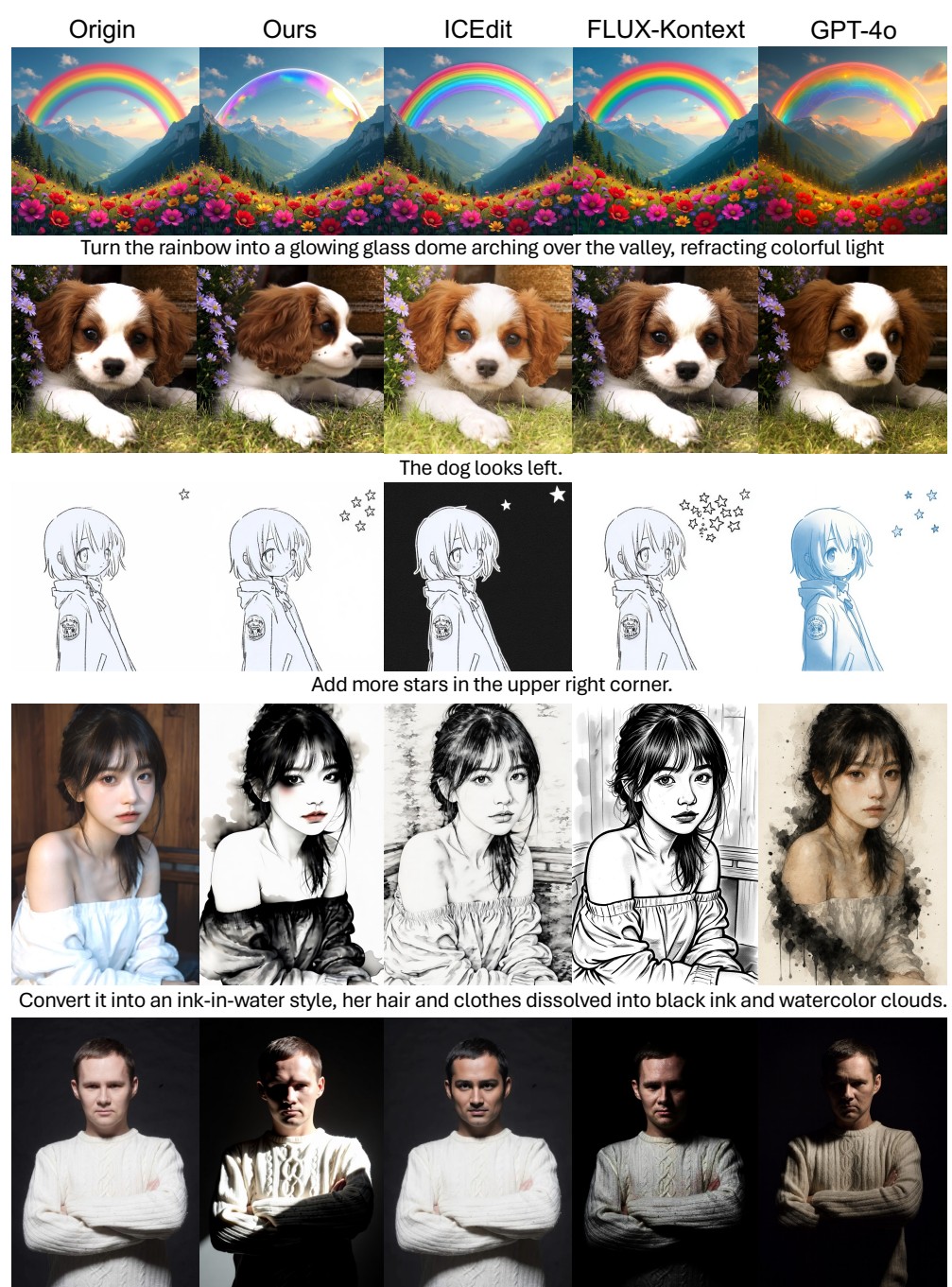

Figure 21: **Qualitative comparison of editing results across different methods**. Our method achieves more faithful instruction following (e.g., rainbow dome, star addition), better identity preservation (e.g., pet and human faces), and stronger background consistency.

# J   MORE QUALITATIVE RESULTS ON TEXT-TO-IMAGE GENERATION

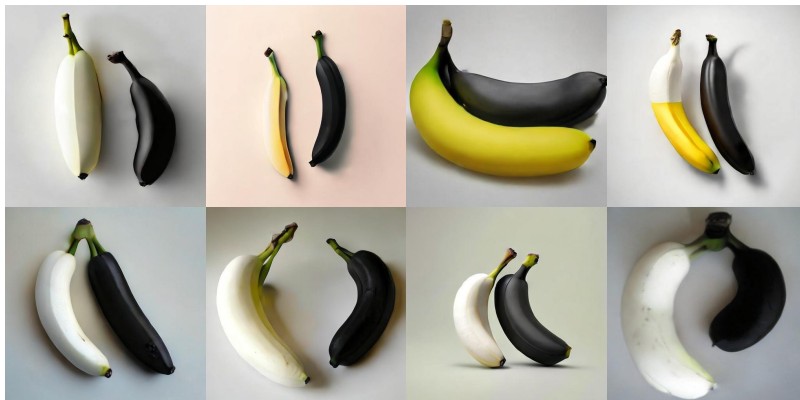

Figure 22: **Uncurated generation results from Harmon-1.5B (top) and post-trained model (bottom)**. Prompt: A white banana and a black banana.

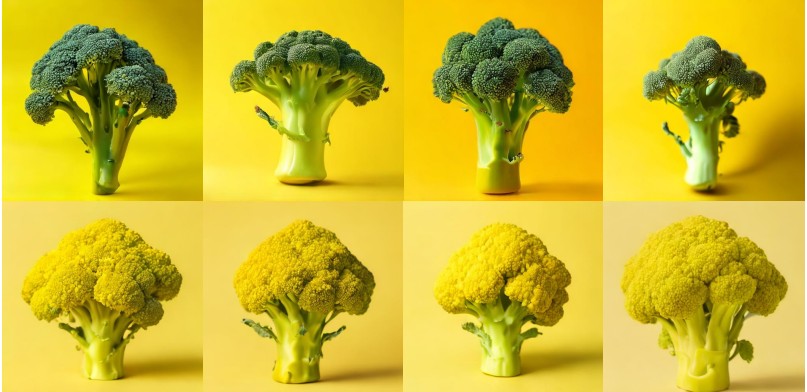

Figure 23: **Uncurated generation results from Harmon-1.5B (top) and post-trained model (bottom)**. Prompt: A photo of a yellow broccoli.

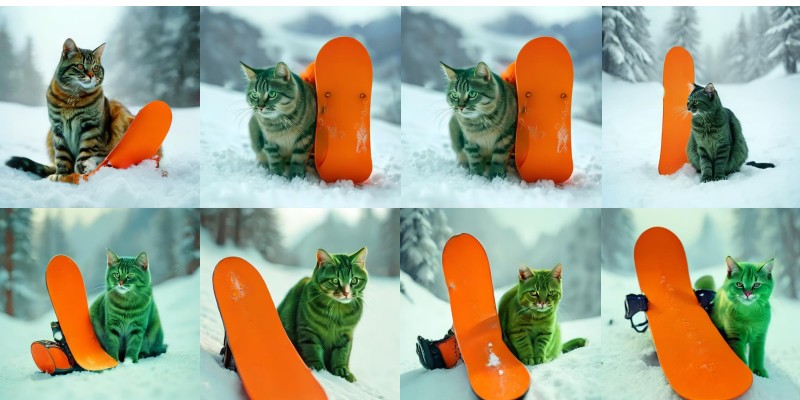

Figure 24: **Uncurated generation results from Harmon-1.5B (top) and post-trained model (bottom)**. Prompt: A photo of an orange snowboard and a green cat.

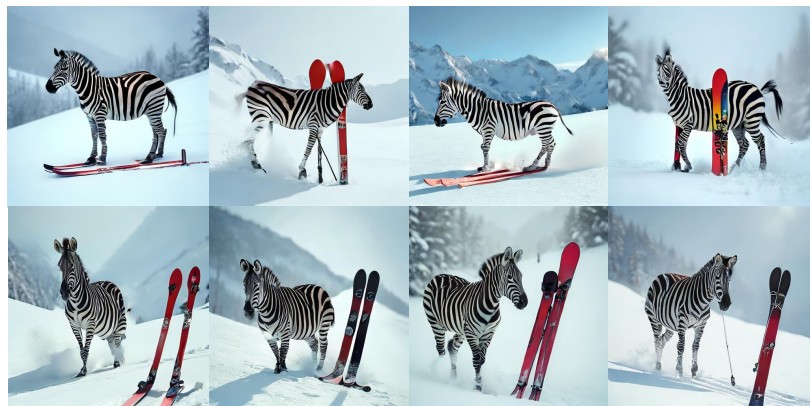

Figure 25: **Uncurated generation results from Harmon-1.5B (top) and post-trained model (bottom)**. Prompt: A photo of a skis right of a zebra.

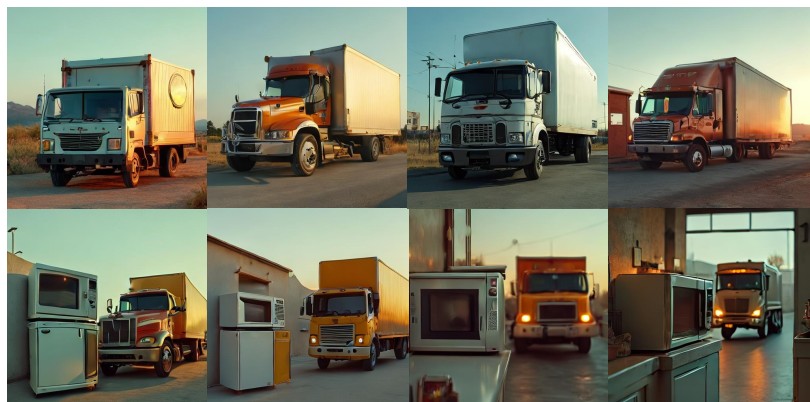

Figure 26: **Uncurated generation results from Harmon-1.5B (top) and post-trained model (bottom)**. Prompt: A photo of a microwave and a truck.

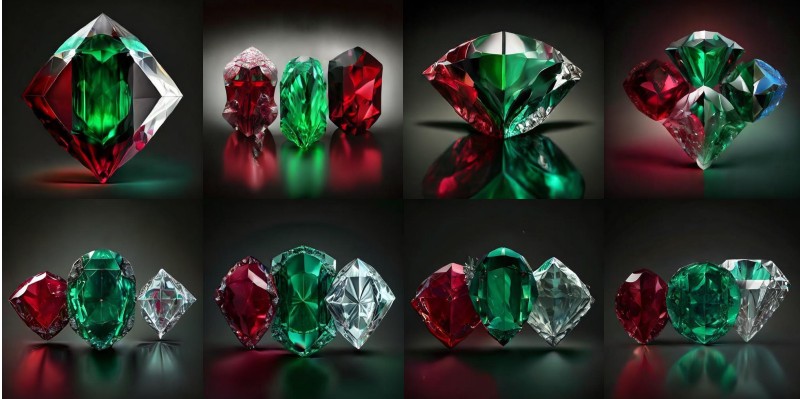

Figure 27: **Uncurated generation results from Harmon-1.5B (top) and post-trained model (bottom)**. Prompt: A diamond on the right, an emerald in the middle, a ruby on the left.

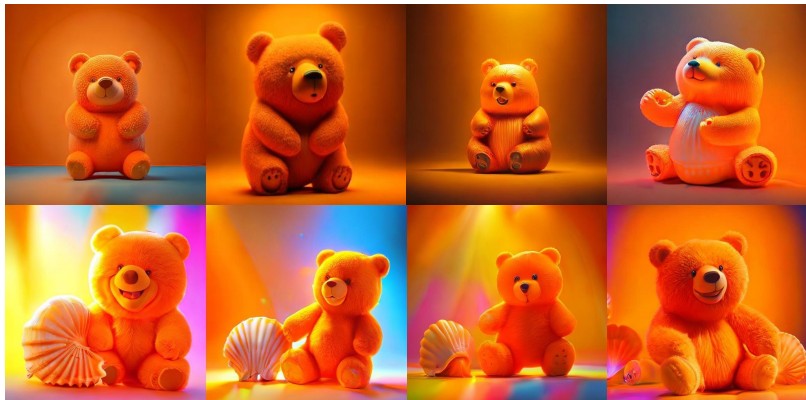

Figure 28: **Uncurated generation results from Harmon-1.5B (top) and post-trained model (bottom)**. Prompt: Cheerful and bright, vibrant lighting. A shell and a bright orange bear in a bright setting.

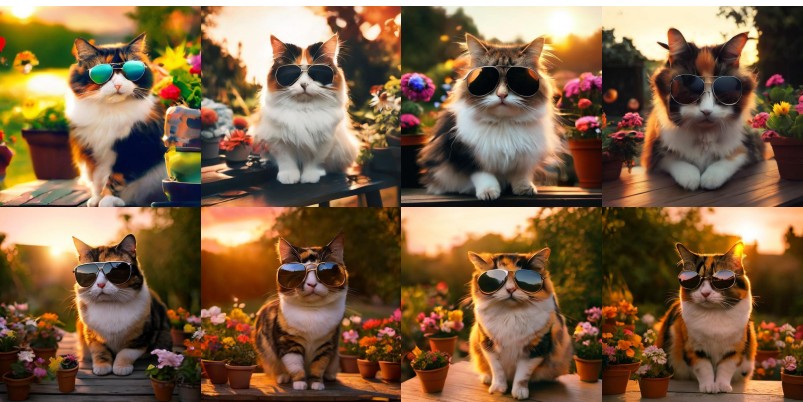

Figure 29: **Uncurated generation results from Harmon-1.5B (top) and post-trained model (bottom)**. Prompt: During the warm glow of a dwindling summer evening, a particular fussy feline with distinctive calico markings is perched atop a garden table. The cat, seemingly indifferent to its surroundings, sports a pair of large, reflective aviator sunglasses that sit comically upon its small, furry face. Around the cat, there are scattered pots of blooming flowers, contributing to the charm of the scene, and in the background, hints of orange and pink skies are visible through the foliage.

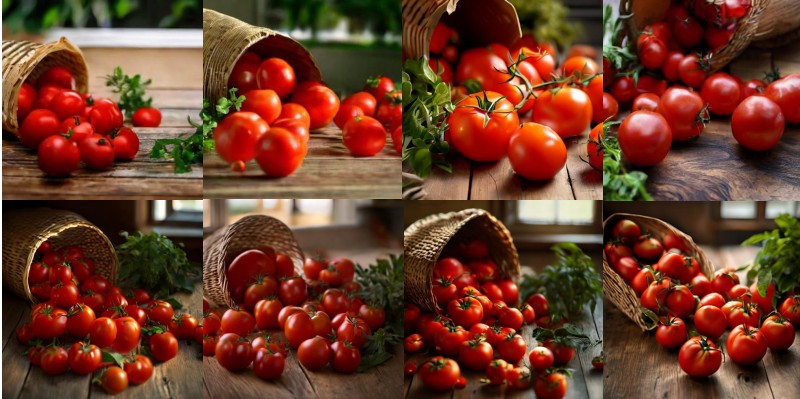

Figure 30: **Uncurated generation results from Harmon-1.5B (top) and post-trained model (bottom)**. Prompt: A vivid scene unfolds where several deep red, perfectly round tomatoes spill from a woven brown basket onto a rustic wooden tabletop. The basket lies on its side as the plump tomatoes scatter across the surface, some touching the dark green leaves of a nearby herb plant. In the background, the blurred outline of an open kitchen window lets in soft, natural light, casting gentle shadows around the fallen produce.

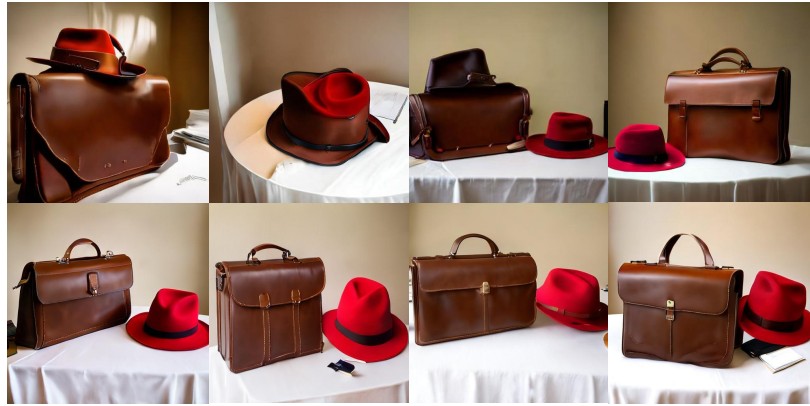

Figure 31: **Uncurated generation results from Harmon-1.5B (top) and post-trained model (bottom)**. Prompt: A polished brown leather briefcase with visible stitching details rests on a white tablecloth, displaying a sense of organization amidst the surrounding environment. Beside the briefcase, a vibrant red fedora hat provides a striking contrast against the pristine table covering. The table, placed in a room with light beige walls, gives an impression of a professional setting with a touch of personal style.

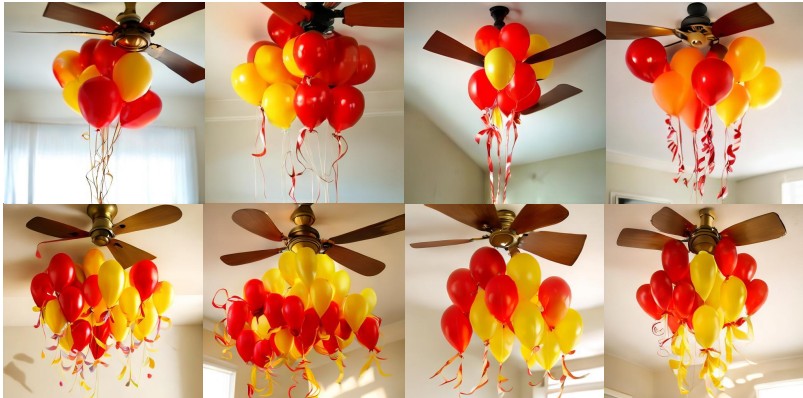

Figure 32: **Uncurated generation results from Harmon-1.5B (top) and post-trained model (bottom)**. Prompt: a festive array of red and yellow balloons tied with curling ribbons, gently bobbing from the breeze of a spinning ceiling fan. The fan has wooden blades and a brass finish, which contrasts with the bright colors of the balloons. The balloons are clustered in a joyful bunch, casting soft shadows on the ceiling above.

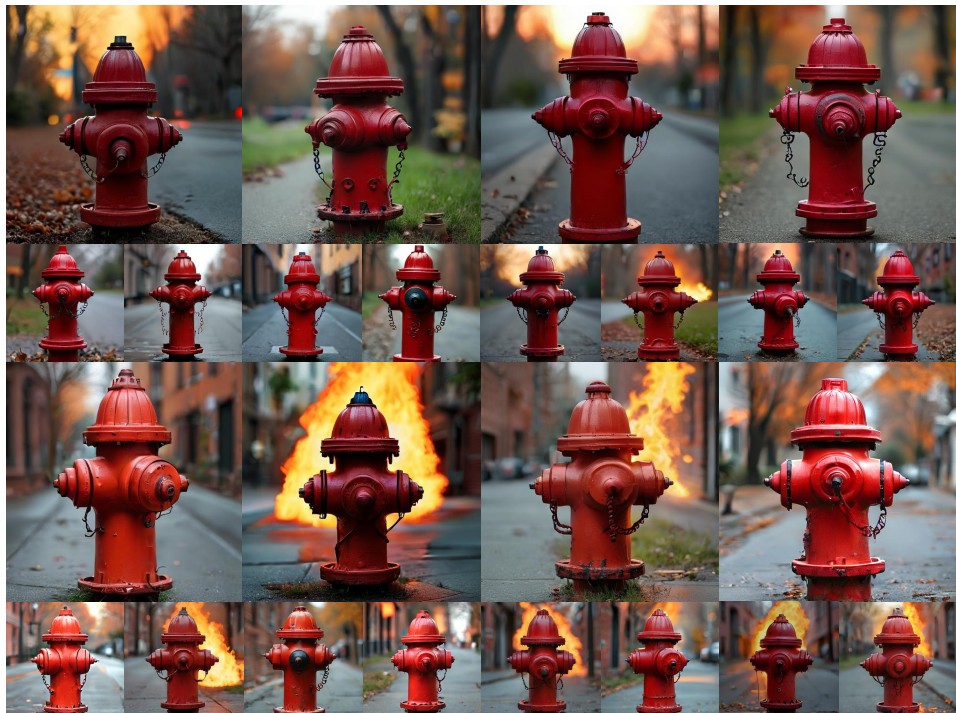

Figure 33: **Uncurated generation results from Harmon-1.5B (top) and post-trained model (bottom)**. Prompt: a photo of a fire hydrant.

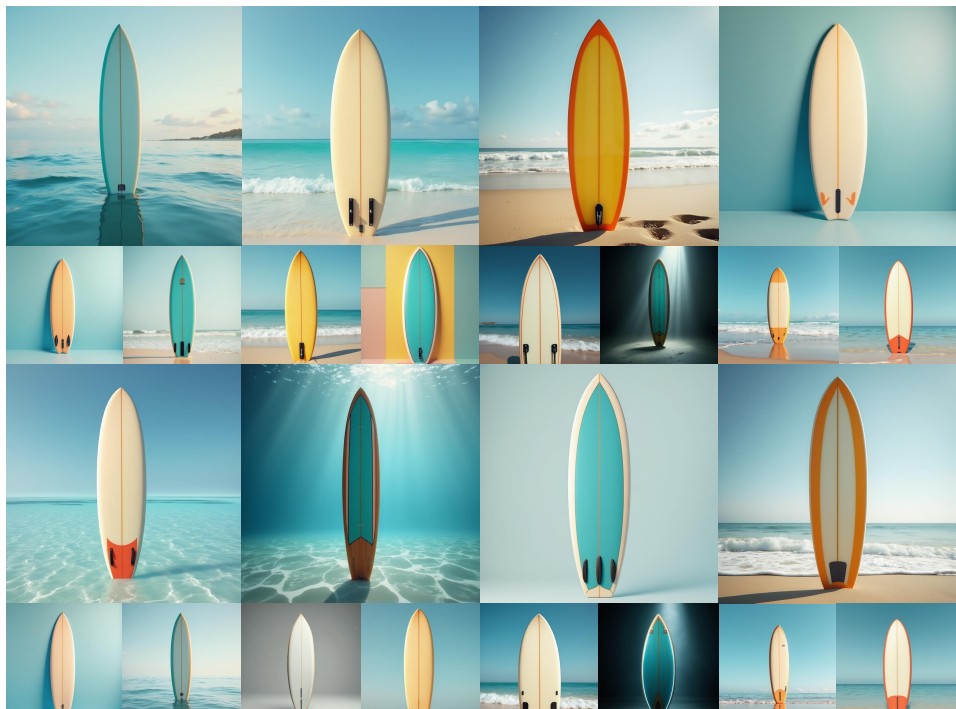

Figure 34: **Uncurated generation results from BAGEL (top) and post-trained model (bottom)**. Prompt: a photo of a surfboard.

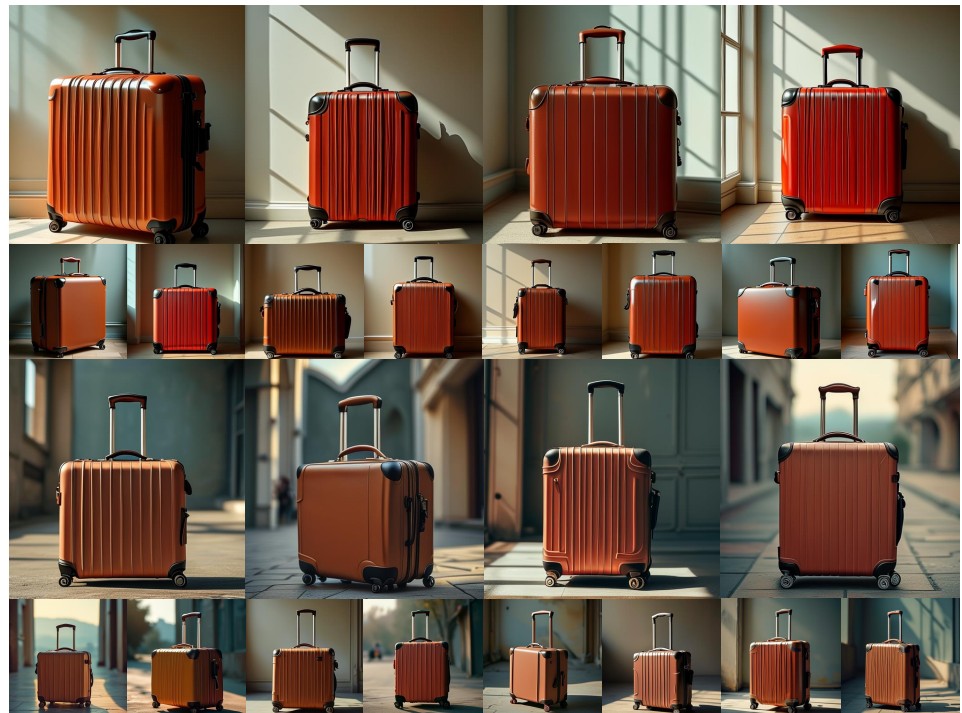

Figure 35: **Uncurated generation results from OpenUni (top) and post-trained model (bottom)**. Prompt: a photo of a suitcase.

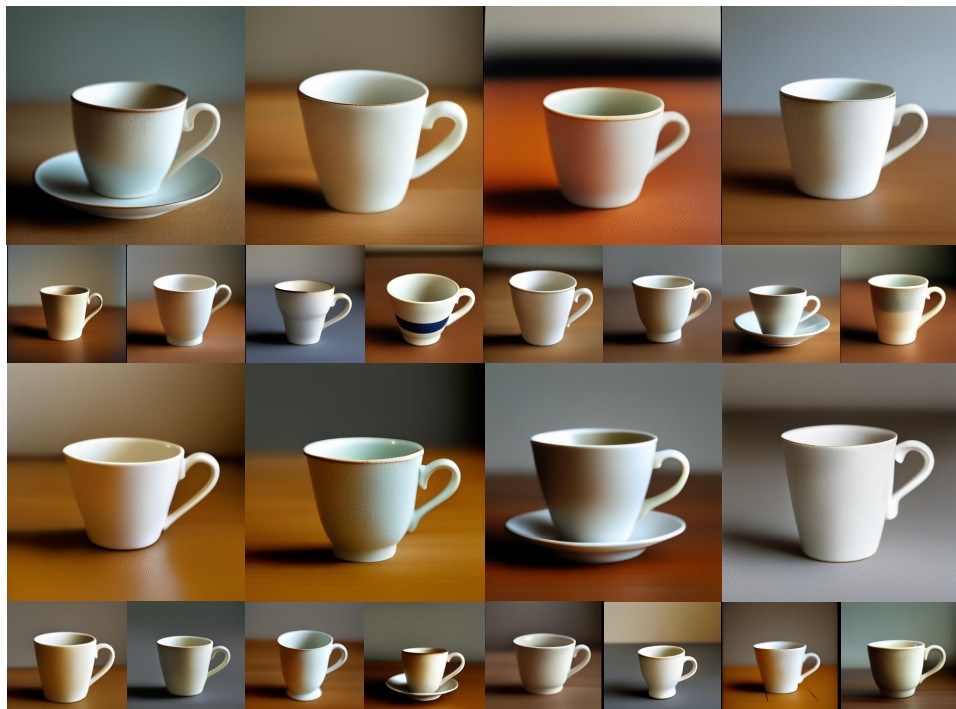

Figure 36: **Uncurated generation results from Show-o-512x512 (top) and post-trained model (bottom)**. Prompt: a photo of a cup.

## K  THE USE OF LARGE LANGUAGE MODELS (LLMS)

During the preparation of this paper, we used large language model (LLM) only as a writing assistant to check grammar, spelling, and to polish the clarity of expression. The LLM was not involved in the research design, experiments, or analysis, and the authors take full responsibility for all content of this work.

