# OpenReview forum: "Reconstruction Alignment Improves Unified Multimodal Models"
_ICLR.cc/2026/Conference — ICLR 2026 Poster_

### Official Review · Reviewer_uKdX · 2025-10-23

**Soundness:** 2
**Presentation:** 3
**Contribution:** 3
**Rating:** 4
**Confidence:** 4

**Summary:**

This paper introduces Reconstruction Alignment (RecA), a lightweight post-training framework designed to improve semantic consistency between the visual understanding and visual generation modules of unified multimodal models (UMMs). The key idea is to use the embeddings from the vision encoder as dense semantic prompts for self-supervised image reconstruction. By enforcing a reconstruction loss, the model explicitly learns to map semantic embeddings to visual outputs, thereby refining the alignment between understanding and generation. RecA is model-agnostic and can be applied to multiple architectures. Extensive experiments on GenEval, DPGBench, and multiple image editing benchmarks demonstrate consistent improvements with minimal computational cost.

**Strengths:**

The proposed reconstruction alignment is conceptually simple yet effective. By reusing visual embeddings as dense prompts, it provides a self-supervised approach to enhance multimodal coherence without relying on additional labeled or distilled data. Across multiple UMM architectures (e.g., Harmon and Show-o), RecA consistently improves the fidelity of text-to-image generation and editing. Furthermore, RecA demonstrates practical efficiency, requiring only 27 A100 GPU hours for deployment.

**Weaknesses:**

- Unified multimodal models (UMMs) aim to integrate both visual understanding and generation within a single architecture. However, RecA is based on a reconstruction task, where the training (I2I) and testing (T2I or I2T) processes differ. It is expected to further clarify how RecA benefits T2I generation tasks as well as visual understanding tasks, especially considering that visual understanding requires both detailed and high-level comprehension.

- Although the work claims to enhance unified multimodal alignment, the quantitative results primarily focus on generation tasks (T2I and editing). Direct evaluation on understanding tasks (e.g., VQA) is needed to substantiate the claim of improving the "unified" aspect of UMMs, as the current results do not fully address this dimension.

- The mechanism by which semantic embeddings improve alignment is primarily intuitive. A deeper representation analysis (e.g., embedding similarity before/after RecA) would strengthen the causal understanding of why RecA works.

- RecA primarily operates as a post-training technique, which inherently limits its scalability.

- Some important technical details are missing. For instance, the paper does not clearly specify where and how the visual embedding is integrated into the model. Additionally, it remains unclear whether hyperparameter tuning is required for each model, which limits the reproducibility of the results.

**Questions:**

Please see weaknesses.

---

> ### Author Response · Authors · 2025-11-22
>
> ### Dear Reviewer uKdX,
>
> Thank you very much for your thoughtful comments and for recognizing the simplicity and effectiveness of our approach. We sincerely appreciate your feedback. We have revised our manuscript based on your comments, below, we address your concerns:
>
> ---
>
> ### **[Q1 & Q2] How RecA benefits T2I generation tasks as well as visual understanding tasks. Direct evaluation on understanding tasks**
>
> **1. Clarification on visual understanding**: Our intention with RecA is to **significantly improve generation quality while *preserving* the model’s existing understanding ability**, rather than to improve standard understanding benchmarks. This is reflected in **Sec. 2.2** and in **Table 4**. Concretely:
>
> * **OpenUni / BAGEL:** The visual understanding module is **frozen**, so its performance naturally remains unchanged.
> * **Shared-parameter UMMs:** We **co-train with an image-to-text loss** ($\lambda_{i2t}=1$), which ensures the original understanding capability is maintained during RecA training.
>
> In short, RecA is designed to enhance generation fidelity **without degrading** visual understanding, which is supported by our experimental results in Table 4 (Section 3.4).
>
> **2. Why I2I reconstruction helps T2I generation**: We agree it is important to clarify this connection. A detailed explanation is provided in **Appendix D (L1184–L1125)** and further discussed in our response to Q3. Briefly:
>
> * The I2I reconstruction objective provide **stronger visual representations**, and
> * improves **cross-modal alignment** used during text-conditioned decoding.
>
> These strengthened components directly translate into better T2I generation fidelity, even though RecA is trained on I2I.
>
> ---
>
> ### **[Q3]: The mechanism by which semantic embeddings improve alignment is primarily intuitive. A deeper representation analysis (e.g., embedding similarity before/after RecA) would strengthen the causal understanding of why RecA works.**
>
> Thank you for the great suggestion! We agree this is an important question to dig into. In response, we added a deeper analysis in **Appendix D**, which highlights two key findings:
>
> * **Visual–Textual Feature Alignment.**
> Using PCA, we found that in the deep transformer layers of a UMM, visual and textual features gradually become *isomorphic* in the semantic space. This means the model can interpret dense visual embeddings almost like language-aligned prompts. A fun example is when the model is shown a rare “pink banana”: internally it maps this to the closest concept it knows (e.g., “pink shoe”), and consistently generates a shoe-like object.
>
> * **Strong Reconstruction–Generation Correlation.**
>   We also observed a strong empirical correlation between reconstruction ability and text-to-image generation quality. For example, when the model struggles to reconstruct a microwave from a cluttered scene, it likewise struggles to generate a microwave from text. This suggests that better reconstruction directly supports better T2I generation.
>
> Together, these findings show that **improving the model’s ability to reconstruct from dense visual prompts naturally strengthens its ability to generate from sparse text prompts**. RecA takes advantage of this connection by using a reconstruction loss to calibrate the model’s semantic space, leading to better generation quality and stronger instruction following. More detailed figures and analysis can be found in **Appendix D**.
>
> Finally, we want to clarify an important detail about our methodology. For models with **decoupled understanding and generation components** (such as **OpenUni** and **BAGEL**), the **visual understanding encoder is frozen** throughout RecA post-training. This means the visual embedding space itself does **not** change; the improvements come from enhancing the **generation module’s ability** to make better use of these fixed embeddings.
>
> ---
>
> ### **[Q4]: RecA primarily operates as a post-training technique, which inherently limits its scalability.**
>
> Thank you for raising this point! While RecA is indeed presented as a **post-training** method, this is actually one of the reasons we are excited about it, **we see this as a practical strength rather than a limitation**. RecA requires **very little compute**, yet provides substantial improvements in generation performance. It can also be applied to **any** pre-trained UMM, regardless of its original training recipe, enabling fast adaptation without any extra external modules, modifying the architecture or retraining the model from scratch.
>
> That said, we fully agree that exploring RecA during **pre-training** could be valuable. Integrating reconstruction alignment earlier, especially with large-scale and diverse data, may further enhance unified multimodal modeling. Due to limited computational resources, we were unable to explore this direction during the short rebuttal period, but we view it as an exciting future work!

---

> > ### Author Response · Authors · 2025-11-22
> >
> > ### **[Q5.1]: Some important technical details are missing. For instance, the paper does not clearly specify where and how the visual embedding is integrated into the model.**
> >
> > Thank you for pointing this out! It is an important aspect of our methodology and we have added relevant descriptions in Sec 2.2 and Appendix C. The process of integrating visual understanding embedding is exactly the same as in image captioning. To be specific, the input for Harmon is:
> >
> >
> > ```
> > <|im_start|>system
> > You are a helpful assistant.<|im_end|>
> > <|im_start|>user
> > <image>
> > Describe this image in detail.<|im_end|>
> > <|im_start|>assistant
> > ```
> >
> >
> > During reconstruction training, the model generates an image based on this prompt just like in image captioning.
> >
> > ---
> > ### **[Q5.2]: Additionally, it remains unclear whether hyperparameter tuning is required for each model, which limits the reproducibility of the results.**
> >
> > Thank you for this important question! RecA requires minimal hyperparameter searching and is easy to reproduce. For most cases, we use **default hyperparameters** (e.g.,learning rate, weight decay) from the original models. adjusting only the batch size to fit our available GPU resources. We find that performance is relatively insensitive to these changes.
> >
> > The only exception is BAGEL, where we slightly reduced the learning rate (from 5e-5 to 4e-5. We observed that 4e-5 yielded marginal gains in image editing benchmarks (ImgEdit and GEdit), leading us to select it as the final setting. However, as shown in the table below, the performance difference between these two settings is **negligible compared to the substantial improvement over the baseline**. This demonstrates that RecA is highly robust to hyperparameter choices. We attribute the need for this minor adjustment to **the limit of computation resources** (fully fine-tuning a 14B model with a limited global batch size of 48), and we believe that with sufficient computational resources (i.e., larger batch sizes), the model would likely exhibit even greater robustness to hyperparameters.
> >
> >
> > | Model Variant  | Learning Rate | GenEval↑ | DPG↑ | ImgEdit↑ | GEdit↑ | Notes   |
> > | ----------------------------- | --------------- | --------- | ---------  | --------- | --------- | --------- |
> > | **BAGEL (baseline, no RecA)** | —| 78.8 |  84.03   | 3.38  | 6.94  | Original model without RecA       |
> > | **BAGEL + RecA** | 5e-5          | 82.0 | **85.35**    | 3.71  |  7.23  | Using BAGEL’s default LR          |
> > | **BAGEL + RecA** | 4e-5          | **82.4** | 85.29  | **3.75**  |  **7.27**  | Smaller BS: smaller LR for stability |
> >
> >
> > Regarding **training steps**, we observe that performance improves rapidly in the early stages and then reaches a plateau, remaining stable with minor fluctuations thereafter. As shown in our added Table 5 and Figure 16, the GenEval scores of post-trained Harmon-1.5B stabilize after a certain number of steps, indicating that RecA training is not sensitive to hyperparameters.
> >
> > | Step     |     0    |    1k    |    2k    |    3k    |    4k    |    5k    |    6k    |    7k    |    8k    |    9k    |    10k   |
> > | :------- | :------: | :------: | :------: | :------: | :------: | :------: | :------: | :------: | :------: | :------: | :------: |
> > | **SFT**  |   72.9   |   74.7   |   73.8   |   75.4   |   74.9   |   74.8   |   74.9   |   74.0   |   73.5   |   73.7   |   73.9   |
> > | **RecA** | **72.9** | **82.1** | **84.1** | **85.0** | **85.0** | **85.7** | **85.7** | **85.0** | **84.7** | **84.6** | **85.0** |
> >
> > Furthermore, all hyperparameters are **listed in Appendix Table 8** for easy reproduction, and we will open-source our code to facilitate academic research and ensure full reproducibility.
> >
> > *Hope our explanation and experiments can address your inquiries. We will integrate all your valuable comments into our revision!*

---

### Official Review · Reviewer_U3Cq · 2025-10-30

**Soundness:** 3
**Presentation:** 3
**Contribution:** 3
**Rating:** 6
**Confidence:** 3

**Summary:**

This paper introduces Reconstruction Alignment (RecA), a simple yet effective post-training method for Unified Multimodal Models (UMMs) that unifies understanding and generation. The key idea is to condition a UMM on its own visual understanding encoder embeddings and optimize it to reconstruct the input image using a self-supervised reconstruction loss.

**Strengths:**

1. Simple yet effective. The key idea is very simple, but surprisingly works well. It yields consistent improvements in image generation and editing performance while being computationally efficient (27 GPU hours).

2. Broad generality: RecA is architecture-agnostic and demonstrated across various UMM families (autoregressive, masked-autoregressive, and diffusion-based). Works across AR, MAR, and AR+Diff UMMs with consistent gains.

3. Good experimental results. Experiments show that a 1.5B-parameter UMM post-trained with RecA outperforms much larger models, achieving GenEval 0.90 and DPGBench 88.15, and improves editing benchmarks such as ImgEdit (3.38→3.75) and GEdit (6.94→7.27).

4. Clarity and reproducibility: Strong visualizations, clear methodology, and detailed appendices.

**Weaknesses:**

1. Limited theoretical grounding: The paper would benefit from more analysis on why reconstruction alignment so effectively bridges understanding–generation gaps (e.g., mutual information or feature alignment metrics).

2. What are the failure cases—does RecA ever degrade diversity or overfit to reconstruction-style artifacts?

**Questions:**

1. Could RecA be extended to cross-modal reconstruction (e.g., text→image→text) for deeper alignment?

2. Could combining RecA with RL-based alignment (e.g., DPO/GRPO) further enhance reasoning alignment?

---

> ### Author Response · Authors · 2025-11-22
>
> ### Dear Reviewer U3Cq,
>
> We sincerely appreciate your recognition of RecA's strong empirical performance and its ability to surpass significantly larger models despite its efficiency. We are also glad you found our approach architecture-agnostic and consistently effective. Thank you for your valuable suggestions! We have carefully considered your feedback and address your questions below:
>
> ---
> ### **[W1] Analysis on why reconstruction alignment so effectively bridges understanding–generation gaps.**
>
> Thank you for your valuable feedback. **We provide a deeper analysis of RecA's mechanism in Appendix D**, with two core findings:
>
> * **Visual–Textual Feature Alignment.**
> Using PCA, we found that in the deep transformer layers of a UMM, visual and textual features gradually become *isomorphic* in the semantic space. This means the model can interpret dense visual embeddings almost like language-aligned prompts. A fun example is when the model is shown a rare “pink banana”: internally it maps this to the closest concept it knows (e.g., “pink shoe”), and consistently generates a shoe-like object.
>
> * **Strong Reconstruction–Generation Correlation.**
> We also observed a strong empirical correlation between reconstruction ability and text-to-image generation quality. For example, when the model struggles to reconstruct a microwave from a cluttered scene, it likewise struggles to generate a microwave from text. This suggests that better reconstruction directly supports better T2I generation.
>
> Together, these findings show that **improving the model’s ability to reconstruct from dense visual prompts naturally strengthens its ability to generate from sparse text prompts**. RecA takes advantage of this connection by using a reconstruction loss to calibrate the model’s semantic space, leading to better generation quality and stronger instruction following. More detailed figures and analysis can be found in **Appendix D**.
>
> ---
> ### **[W2] What are the failure cases—does RecA ever degrade diversity or overfit to reconstruction-style artifacts?**
>
> Thank you for this important question. We have not observed any degradation in image quality or diversity.
>
> - **Regarding image quality**, we have not observed any reconstruction-style artifacts. The model's text-to-image generation capability remains robust, and the consistent performance improvements on benchmarks that evaluate image quality, such as WISE and ImgEdit, further validate this point.
>
> - **Regarding diversity**, we have not observed noticeable reduction. To better illustrate this, we have included additional qualitative results in **Appendix J**, where we generate 12 images from a single prompt. As shown, the generated images exhibit a high degree of diversity in style, composition, and detail, demonstrating that RecA does not harm the model's generative diversity. We will open-source all our code and models to facilitate further verification.
>
> ---
>
> ### **[Q1] Could RecA be extended to cross-modal reconstruction (e.g., text→image→text) for deeper alignment?**
>
> Absolutely! There are already some similar works exploring this cross-modal reconstruction. UniRL [3] and SRUM [4] first use text prompts to generate images and then let the UMM do VQA tasks to obtain reward signals. UAE [5] lets a UMM describe a given image, feeds the description back to the model and generates another image and finally gets reward signals from differences between two images. These works have already verified your suggestion, and we could extend RecA by similarly imitating UAE:
>
> - Use a text prompt *c* to generate an image *I*.
>
> - Feed *I* back to the UMM and generate another image description *c’*.
>
> - Use a reward model to compute the similarity between *c* and *c’*.
>
> Furthermore, your questions inspire us that we can extend RecA to other modalities like voice, speech and video. Based on the insight from RecA, there are so many directions that we can explore.
>
> References
> > [3] UniRL: Self-Improving Unified Multimodal Models via Supervised and Reinforcement Learning, Mao et al, Arxiv 2025.
> >
> > [4] SRUM: Fine-Grained Self-Rewarding for Unified Multimodal Models, Jin et al, Arxiv 2025.
> >
> > [5] Unified Multimodal Model as Auto-Encoder, Yan et al, Arxiv 2025.

---

> > ### Author Response · Authors · 2025-11-22
> >
> > ### **[Q4] Could combining RecA with RL-based alignment (e.g., DPO/GRPO) further enhance reasoning alignment?**
> >
> > Yes! RL optimizes for preference feedback and reasoning, while RecA optimizes for dense semantic reconstruction. These orthogonal objectives can be combined without conflict.
> >
> > First, RecA can seamlessly integrate with models' CoT process, further enhance UMM's reasoning-based generation. We report the WISE scores of BAGEL + RecA + CoT in the table below:
> >
> > |Model|Cultural|Time|Space|Biology|Physics|Chemistry|Overall|
> > |-|-|-|-|-|-|-|-|
> > |BAGEL w/ CoT|0.73|0.67|0.77|0.64|0.76|0.65|0.70|
> > |BAGEL-RecA w/ CoT|**0.75**|**0.68**|**0.81**|**0.66**|**0.78**|**0.68**|**0.73**|
> >
> > The results show that RecA consistently improves reasoning performance across all WISE subtasks when combined with CoT, with an overall improvement of +0.03 (0.70→0.73).
> >
> > Second, we post-trained RL-tuned UMMs with RecA. There are already some RL based work that trains Show-o and Janus, and we choose Mask-GRPO [6] and HermesFlow [7] as our baseline. The results are shown in the table below:
> >
> > |Model|GenEval|DPG|
> > |-|-|-|
> > |Show-o (VQGAN) |68.7|83.63|
> > |HermesFlow|69.1|83.92|
> > |HermesFlow + RecA|71.4|**85.77**|
> > |Mask-GRPO|70.3|85.13|
> > |Mask-GRPO + RecA|**72.1**|85.71|
> >
> > Note that HermesFlow and Mask-GRPO use Show-o's VQGAN variant as their baseline, but we use Show-o's CLIP variant in our main result. We have added some exploration and experiment results of Show-o's VQGAN variant in Appendix F.2.
> >
> > References
> > > [6] Hermesflow: Seamlessly closing the gap in multimodal understanding and generation, Yang et al, NeurIPS 2025.
> > >
> > > [7] Reinforcement Learning Meets Masked Generative Models: Mask-GRPO for Text-to-Image Generation, Luo et al, NeurIPS 2025.
> >
> >
> >
> > ***Hope our explanation and experiments can address your inquiries. We will integrate all your valuable comments into our revision!***

---

### Official Review · Reviewer_xRHK · 2025-11-01

**Soundness:** 3
**Presentation:** 3
**Contribution:** 3
**Rating:** 8
**Confidence:** 4

**Summary:**

This paper presents Reconstruction Alignment (RecA), a simple yet remarkably effective post-training technique for unified multimodal models (UMMs). RecA addresses the limitation of sparse text-caption supervision by leveraging dense supervision from visual encoder embeddings, training the model to reconstruct the original image. This process realigns the model's understanding and generation capabilities without requiring any additional labeled data. Extensive experiments across diverse UMM architectures (autoregressive, hybrid, diffusion-based) demonstrate consistent performance gains on both generation (GenEval, DPGBench) and editing (ImgEdit, GEdit) benchmarks. Notably, the method is computationally lightweight.

**Strengths:**

The proposed method is characterized by its simplicity and broad applicability, ensuring easy implementation with low computational overhead. It demonstrates strong empirical performance, evinced by consistent improvements across four distinct model families. The study maintains rigorous experimental standards, including fair comparisons and thorough ablation studies (e.g., SFT vs. RecA, encoder choice, resolution). The approach has high practical relevance as a plug-and-play component for future UMM pipelines, and its benefits are clearly illustrated through intuitive figures and examples

**Weaknesses:**

1. The heavy reliance on simple benchmarks (GenEval and DPGBench) raises questions about true generalization. The absence of tests for real-world prompt fidelity and compositional reasoning

**Questions:**

1. Could the authors provide an analysis, perhaps using feature attribution methods, to identify where RecA contributes most significantly, such as spatial layout, geometry, or fine-grained attributes?
2. Could applying RecA iteratively—through multiple cycles of reconstruction—lead to further improvements in the output?
3. How does RecA perform when given open-ended or abstract prompts for which no concrete visual reference exists?
4. Is there a quantifiable trade-off between the fidelity of the reconstructions and the diversity of the generated samples?

---

> ### Author Response · Authors · 2025-11-22
>
> ### Dear Reviewer xRHK,
>
> We are grateful for your positive assessment of our work, particularly your appreciation of RecA's strong empirical performance and its practical value as a plug-and-play component for future UMM pipelines. We have incorporated your suggestions into our revision and provide detailed responses to your concerns below:
>
> ---
> ### **[W1] The heavy reliance on simple benchmarks (GenEval and DPGBench) raises questions about true generalization. The absence of tests for real-world prompt fidelity and compositional reasoning.**
>
> Thank you for your valuable feedback. We strictly follow the evaluation from previous work [1,2], which use GenEval and DPGBench as standard benchmarks. While GenEval prompts may appear simple, DPGBench actually has a comprehensive range of complex attributes and conditions (including fine-grained compositional reasoning), which we believe adequately demonstrates RecA's generalization capability.
>
> To directly address your concern regarding "simple benchmarks" and "compositional reasoning," we conducted additional rigorous evaluations on T2I-CompBench [3]. Unlike GenEval, it comprises 2,670 curated samples across **10 distinct dimensions**, specifically targeting high-difficulty tasks like 3D spatial relationships, complex attribute binding, and complex scene composition. As detailed in Appendix F.4, RecA delivers substantial and consistent gains, demonstrating its true generalization ability.
>
> ### **OpenUni**
>
> |Category|Color|Shape|Texture|Spatial|Non-Spatial|Numeracy|3D Spatial|Complex|Overall|
> |:---|:--:|:--:|:--:|:--:|:--:|:--:|:--:|:--:|:--:|
> |**OpenUni**|87.47|75.22|82.87|78.29|83.10|64.44|68.60|82.82|78.84|
> |**OpenUni-RecA**|**92.50**|**85.60**|**89.65**|**86.30**|**87.18**|**75.97**|**84.35**|**87.79**|**86.49**|
> |*vs. baseline*|$\textcolor{ForestGreen}{+5.03}$|$\textcolor{ForestGreen}{+10.38}$|$\textcolor{ForestGreen}{+6.78}$|$\textcolor{ForestGreen}{+8.01}$|$\textcolor{ForestGreen}{+4.08}$|$\textcolor{ForestGreen}{+11.53}$|$\textcolor{ForestGreen}{+15.75}$|$\textcolor{ForestGreen}{+4.97}$|$\textcolor{ForestGreen}{+7.65}$|
>
> ### **Harmon**
>
> |Category|Color|Shape|Texture|Spatial|Non-Spatial|Numeracy|3D Spatial|Complex|Overall|
> |:---|:--:|:--:|:--:|:--:|:--:|:--:|:--:|:--:|:--:|
> |**Harmon**|89.55|79.09|86.28|84.98|82.77|69.19|75.58|82.05|81.36|
> |**Harmon-RecA**|**93.72**|**84.32**|**91.37**|**92.37**|**87.12**|**79.78**|**86.08**|**88.51**|**88.03**|
> |*vs. baseline*|$\textcolor{ForestGreen}{+4.17}$|$\textcolor{ForestGreen}{+5.23}$|$\textcolor{ForestGreen}{+5.09}$|$\textcolor{ForestGreen}{+7.39}$|$\textcolor{ForestGreen}{+4.35}$|$\textcolor{ForestGreen}{+10.59}$|$\textcolor{ForestGreen}{+10.50}$|$\textcolor{ForestGreen}{+6.46}$|$\textcolor{ForestGreen}{+6.67}$|
>
> ### **Show-o**
>
> |Category|Color|Shape|Texture|Spatial|Non-Spatial|Numeracy|3D Spatial|Complex|Overall|
> |:---|:--:|:--:|:--:|:--:|:--:|:--:|:--:|:--:|:--:|
> |**Show-o**|88.60|77.68|85.23|85.75|**85.38**|71.39|75.38|82.73|82.12|
> |**Show-o-RecA**|**90.88**|**82.22**|**88.37**|**87.28**|85.13|**74.42**|**81.42**|**85.07**|**84.47**|
> |*vs. baseline*|$\textcolor{ForestGreen}{+2.28}$|$\textcolor{ForestGreen}{+4.54}$|$\textcolor{ForestGreen}{+3.14}$|$\textcolor{ForestGreen}{+1.53}$|-0.25|$\textcolor{ForestGreen}{+3.03}$|$\textcolor{ForestGreen}{+6.04}$|$\textcolor{ForestGreen}{+2.34}$|$\textcolor{ForestGreen}{+2.35}$|
>
> ### **BAGEL**
>
> |Category|Color|Shape|Texture|Spatial|Non-Spatial|Numeracy|3D Spatial|Complex|Overall|
> |:---|:--:|:--:|:--:|:--:|:--:|:--:|:--:|:--:|:--:|
> |**BAGEL**|93.05|84.25|88.95|88.42|88.27|79.43|81.47|87.85|86.74|
> |**BAGEL-RecA**|**93.62**|**85.72**|**90.58**|**88.57**|**88.80**|**81.88**|**83.50**|**88.74**|**87.89**|
> |*vs. baseline*|$\textcolor{ForestGreen}{+0.57}$|$\textcolor{ForestGreen}{+1.47}$|$\textcolor{ForestGreen}{+1.63}$|$\textcolor{ForestGreen}{+0.15}$|$\textcolor{ForestGreen}{+0.53}$|$\textcolor{ForestGreen}{+2.45}$|$\textcolor{ForestGreen}{+2.03}$|$\textcolor{ForestGreen}{+0.89}$|$\textcolor{ForestGreen}{+1.15}$|
>
> Complete results and analysis can be found in Appendix F.4, demonstrating that RecA effectively enhances compositional reasoning and fine-grained attribute binding in complex scenarios.
>
> Regarding "real-world prompt fidelity," we have already evaluated the WISE benchmark with 1,000 real-world prompts like "Einstein's most favorite instrument", where **no concrete visual reference exists**. As shown in Table 2, RecA improves Harmon from 0.41→0.50 and OpenUni from 0.43→0.54, while BAGEL shows minimal improvement and Show-o shows no improvement.
>
> References:
> > [1] Hermesflow: Seamlessly closing the gap in multimodal understanding and generation, Yang et al, NeurIPS 2025.
> >
> > [2] Vision as a Dialect: Unifying Visual Understanding and Generation via Text-Aligned Representations, Han et al, NeurIPS 2025.
> >
> > [3] T2I-CompBench: A Comprehensive Benchmark for Open-world Compositional Text-to-image Generation, Huang et al, NeurIPS 2023.

---

> > ### Author Response · Authors · 2025-11-22
> >
> > ### **[Q1] Analysis to identify where RecA contributes most significantly (e.g., spatial layout, geometry, or fine-grained attributes).**
> >
> > Thank you for this insightful suggestion. Our extra T2I-Compbench's results results already provide strong evidence and analysis regarding RecA's contributions to specific aspects like fine-grained attributes and spatial reasoning.
> >
> >
> > As detailed in our response to Weakness 1 and the analysis in Appendix F.4, RecA demonstrates substantial gains on specific sub-tasks within GenEval and T2I-Compbench that directly correspond to the areas you mentioned:
> >
> >
> > - **Fine-grained Attributes** (e.g., color, texture, geometric shape): The "Color" (GenEval), "Color Attributes" (GenEval), "Color" (T2I-CompBench), "Shape" (T2I-CompBench), and "Texture" (T2I-CompBench) sub-tasks all show significant improvement, indicating RecA generally enhances the model's ability to bind attributes to objects correctly.
> >
> >
> > - **Spatial Layout**: The "Position" (GenEval), "Spatial" (T2I-CompBench), and "3D Spatial" (T2I-CompBench) sub-tasks also see a major performance boost, confirming that RecA improves the model's generation capability of spatial relationships.
> >
> > In contrast, the slight improvement in the "Counting" (GenEval) sub-task and the unstable performance on the WISE benchmark (which evaluates world knowledge and logical reasoning) suggest that RecA's impact on these higher-level reasoning abilities is less consistent. This indicates that while RecA excels at enhancing visual and spatial fidelity, improving abstract reasoning capabilities remains a more challenging open problem.
> >
> > ---
> >
> > ### **[Q2] Could applying RecA iteratively—through multiple cycles of reconstruction—lead to further improvements in the output?**
> >
> > Thank you for this creative suggestion! We understand your question as: (1) generate images with the baseline model, (2) train RecA on these generated images, (3) generate new images with the RecA-enhanced model, (4) repeat the process iteratively. This represents an intriguing self-improvement paradigm.
> >
> >
> > We have conducted iterative RecA experiments to test this hypothesis before. **The first RecA iteration successfully improves the model using its own generated images, but the second iteration shows no further gains.** The diminishing returns suggest that once RecA aligns the model's understanding and generation capabilities, the reconstruction objective reaches a performance plateau. Further iterations do not introduce new supervisory signals since the model is already well-aligned. So how to make the UMM continuously improve through iterations remains an open question.
> >
> >
> > ---
> >
> > ### **[Q3] How does RecA perform when given open-ended or abstract prompts for which no concrete visual reference exists?**
> >
> > Please see our reply for Weakness 1.
> >
> > ---
> >
> > ### **[Q4] Is there a quantifiable trade-off between the fidelity of the reconstructions and the diversity of the generated samples?**
> >
> > Thank you for this insightful question. We have not observed a decrease in generative diversity. Our qualitative results in Appendix J support this. It is important to note that the field currently lacks a consensus on a reliable quantitative metric for evaluating diversity in text-to-image models, making qualitative assessment an unexplored field.
> >
> >
> > Furthermore, as shown in our newly added training step analysis (Table 5 and Figure 16), with more training steps and a better reconstruction ability, the model's text-to-image performance stabilizes without degradation. This indicates that, within our training paradigm, better reconstruction fidelity positively correlates with T2I performance rather than creating a trade-off. However, we concede that if the reconstruction training were to be extended significantly (e.g., for millions of steps), it is possible that the balance could shift and a trade-off might emerge.
> >
> > ***Hope our explanation and experiments can address your inquiries. We will integrate all your valuable comments into our revision!***

---

### Official Review · Reviewer_ccV6 · 2025-11-03

**Soundness:** 4
**Presentation:** 4
**Contribution:** 4
**Rating:** 8
**Confidence:** 5

**Summary:**

This paper identified an important misalignment and information gap during the training of image generation (text to image): text input provides only sparse visual information while the model learns to fine-grained visual details. The sparsity holds even when the prompt is completed with hundreds of words. In the context of a unified multimodal model (UMM), the mapping from visual features to language space gives rise to a new source of dense supervision for image generation, i.e., the semantic embeddings of the target image itself. Built on this insight, the authors proposed a post-training technique named Reconstruction Alignment (RecA), where images are abstracted by a visual encoder and mapped to the input space of LLM, then the generation part of the UMM reconstructs the input image. This enables the learning of structural and low-level details of images that may be rare in image captions. RecA was experimented on various types of UMM and brought consistent gains for both image generation and editing.

**Strengths:**

1. The core idea of the paper, i.e, dense conditioning for image generation, is intuitive and well-motivated. It has been a long-standing issue in multimodal learning that generating visual details is much easier than perceiving them. The authors' perspective on the information density of condition signals is insightful.

2. And the solution, a simple reconstructive supervision, is widely applicable to various UMM paradigms and is effective across different benchmarks. Particularly, the results on GenEval are exceptional, without relying on gpt-distilled data samples.

3. The paper is well written and easy to follow.

**Weaknesses:**

1. The proposed method seems limited to the post-training stage. It is unclear whether ReCA will also benefit the pretraining of image generation models or unified models.

2. More reconstruction results are expected for all the models studied. I believe the reviewers and readers would be interested in how well the images can be reconstructed.

3. It is unclear how the performance would be affected by the training images, e.g., the scale of the data, visual quality, real data v.s., synthetic data.

4. A factual error: Show-o is not an AR model for image generation. Instead, it is a mix of AR for understanding and discrete diffusion for generation.

**Questions:**

1. The improvement on Harmon is exceptionally larger than on other models. Can the authors provide more insight? Would it be due to the shared visual encoder or because Harmon mainly uses short captions for training, thus leaving a large space for improvement with additional dense supervision?

2. The performance gains on GenEval are crystal clear. However, GenEval prompts are all short, and the evaluation mainly focuses on high-level image understanding like object detection and counting. Why would the dense supervision be so effective?

---

> ### Author Response · Authors · 2025-11-22
>
> ### Dear Reviewer ccV6,
>
> Thank you for your encouraging feedback! We are delighted that you found our core concept of dense conditioning intuitive and well-motivated, and that you appreciated the wide applicability of our method. We also value your recognition of our outstanding results on GenEval. We have addressed your comments point-by-point below:
>
> ---
>
> ### **[W1] The proposed method seems limited to the post-training stage. It is unclear whether ReCA will also benefit the pretraining.**
>
> Thank you for this concern! We acknowledge that exploring RecA at the pre-training stage could be valuable and potentially further enhance UMMs' generation capabilities with high-quality, large-scale data. However, verifying this would require training models from scratch with massive computational resources, which is currently beyond our capability.
>
>
> We believe that being a post-training method is actually one of RecA's key advantages: it enables efficient and accessible improvements to any pre-trained UMM without costly retraining from scratch. We plan to explore pre-training stage integration as important future work when resources permit.
>
>
> ---
>
> ### **[W2] More reconstruction results are expected for all the models studied.**
>
> Thank you for this valuable suggestion! We have added comprehensive reconstruction visualizations in **Appendix F.7** with two complementary figures:
>
>
> - **Figure 17** shows Harmon's reconstruction progression across different training timesteps, demonstrating how reconstruction capability gradually improves to high-fidelity reconstruction results.
>
>
> - **Figure 18** compares reconstruction quality before and after RecA post-training for all evaluated models (Show-o, Harmon, OpenUni, BAGEL).
>
>
> ---
>
> ### **[W3] It is unclear how the performance would be affected by the training images (e.g., scale, visual quality, real vs. synthetic).**
>
> Thank you for this important question! We have conducted comprehensive ablation studies to evaluate RecA's robustness across different data scales and sources.
>
>
> **Data Scale Analysis:** We evaluated Harmon-1.5B with varying amounts of MidjourneyV6 data:
>
>
> | Training Data | Data Scale | GenEval  | DPG  |
> | ------------- | ---------- | -------- | --------- |
> | MidjourneyV6  | 10k        | 84.6     | 86.82     |
> | MidjourneyV6  | 50k        | 85.3     | 87.06     |
> | MidjourneyV6  | **240k**   | **85.7** | **87.21** |
>
>
> The results show consistent performance improvements with more data.
>
>
> **Data Source Robustness:** As reported in Table 5, we have evaluated RecA with different data sources like BLIP3o-60k. To strengthen our conclusion, we conducted additional experiments using JourneyDB (synthetic data generated by a lower version of Midjourney) and COCO2014 (real world data, with relatively low visual quality images):
>
>
> | Data Source  | Quality | GenEval  | DPG  |
> | ------------ | ------- | -------- | --------- |
> | JourneyDB    | Medium  | 85.2     | 86.71     |
> | BLIP3o-60k   | High    | 85.2     | 86.50     |
> | MidjourneyV6 | High    | **85.7** | **87.21** |
> | COCO2014     | Low     | 85.3     | 86.82     |
>
>
>
> The results are aligned with what we claimed in Sec 3.5 that RecA maintains relatively stable and robust performance across different data sources, regardless of whether they are synthetic or real-world datasets and different visual qualities. These new experimental results have been added to **Appendix F.8**.
>
> ---
>
> ### **[W4] A factual error: Show-o is not an AR model for image generation.**
>
> Thank you for pointing this out! Show-o is based on the MASKGIT [1] paradigm. We previously categorized it as an AR model just because it uses cross-entropy loss applied on LLM. We are trying to prove that RecA can work on both diffusion loss and cross-entropy loss.
>
>
> The classification of UMMs is a controversial problem, and in a survey [2] there are even 8 types of UMMs. So to clarify and avoid confusion, we have reclassified UMMs in our revised paper based on their generation mechanisms: (1) **Discrete UMM**, (2) **Continuous UMM**, (3) **MAR UMM**, which better reflects the underlying paradigm.
>
>
> If you have further suggestions on UMM classification, we would be happy to discuss and refine it.
>
> References
> > [1] MaskGIT: Masked Generative Image Transformer, Chang et al, CVPR 2022.
> >
> > [2] Unified Multimodal Understanding and Generation Models: Advances, Challenges, and Opportunities, Zhang et al, Arxiv 2025.

---

> ### Author Response · Authors · 2025-11-22
>
> ### **[Q1] The improvement on Harmon is exceptionally larger than on other models. Can the authors provide more insight?**
>
> Thank you for this excellent question! To investigate whether the shared visual encoder (MAE) is the key factor, we conducted an ablation study by freezing the MAE encoder during RecA training. The results are:
>
>
> | Setting | GenEval | DPG |
> |---------|---------|----------|
> | Frozen MAE encoder | 85.4 | 87.05 |
> | Trainable MAE encoder | **85.7** | **87.21** |
>
>
> The minimal difference (+0.3 GenEval, +0.16 DPG) suggests that **simultaneously tuning shared-encoder is not the primary source of Harmon's exceptional improvement**. Instead, we attribute the gains to the reconstruction capability learning itself.
>
>
> To conclusively identify which architectural component contributes most (shared-encoder, MAR decoder, etc.) would require extensive **from-scratch** ablation studies, comparing variants with/without shared-encoders or with/without MAR decoders.
>
>
> Regarding your hypothesis about training captions: Harmon's training data (JourneyDB) actually contains numerous dense prompts, as stated in its original paper. Therefore, we believe Harmon's exceptional gains stem from its architectural advantages. Specifically, Harmon's unique design may be particularly well-suited for RecA.
>
> ---
>
> ### **[Q2] Why would the dense supervision be so effective given GenEval prompts are short and focus on high-level understanding?**
>
> Thank you for this insightful question! The key lies in understanding what images contain versus what text captions describe. While GenEval prompts appear simple (e.g., "a red apple and a blue banana"), images themselves are information-rich and encode abundant visual concepts that are often overlooked or forgotten during biased text-to-image supervision.
>
>
> **Why is dense supervision effective?** Consider a landscape photograph: it implicitly contains rich visual concepts like `cloudy sky`, `brick-textured walls`, `a cow is left of the house`, etc. These visual concepts could be rarely described in typical captions, leading models to develop **biased correlation** (e.g., broccoli → green, banana → yellow, as shown in Figure 3).
>
>
> Visual understanding embeddings encode not just individual concepts but their relationships: how objects are positioned, how colors are distributed, how shapes interact. By reconstructing from visual understanding embeddings, the model is forced to learn these concepts that were ignored in caption-based training. RecA's dense supervision directly trains the model on these compositional relationships that sparse text fails to capture.
>
>
> GenEval fundamentally tests compositional generation (e.g., combining "red" + "apple" + "blue" + "banana"). That is why even cases with simple prompts can benefit from RecA's dense supervision. Furthermore, we have provided a detailed mechanistic analysis in **Appendix D**, where you can find more visualizations and insights on why RecA works so effectively.
>
>
> ***Hope our explanation and experiments can address your inquiries. We will integrate all your valuable comments into our revision!***

---

> ### Comment · Reviewer_ccV6 · 2025-11-24
> **Comprehensive Study and Analysis**
>
> A big thanks to the authors for the exhaustive responses and clarifications. The absence of study on pre-training and lack of consistent gains on understanding tasks remain as limitations but do not constitute reasons for rejection. The reviewer insists on a clear acceptance of this paper.

---

> > ### Author Response · Authors · 2025-11-24
> >
> > Thank you for your prompt response and positive feedback. Your recognition is both encouraging and deeply appreciated, and we will continue to investigate the effect of RecA in the pre-training stage and explore how to further enhance UMM’s understanding capabilities.
> >
> > Your insightful questions and suggestions have been truly valuable in improving our work. Please feel free to let us know if you have any additional comments or further suggestions.

---

### Author Response · Authors · 2025-11-22
**Global Rebuttal**

### **We sincerely thank the reviewers for their thoughtful comments and encouraging feedback!**

In this paper, we introduce RecA, a new self-supervised post-training approach to enhance generation ability of Unified Multimodal Model (UMM) without use of any text caption and external models. We are encouraged by the recognition of our contributions on:


- **Intuitive and Effective**: Reviewer ccV6 acknowledges our main concept of dense conditioning as ***“intuitive and well-motivated,”*** giving a ***“insightful” view point of information density***. Reviewer xRHK and Reviewer U3Cq find that RecA has ***“strong empirical performance” and “surprisingly works well.”*** Reviewer uKdX further highlights the benefit of our self-supervised method improving performance ***“without consuming any extra labeled or distilled data.”***


- **Promising Generality and Applicability**: Reviewer U3Cq and Reviewer uKdX like that RecA is ***“architecture-agnostic” and “consistently improves the fidelity”***. Reviewer ccV6 remarks that the technique is “widely applicable” and Reviewer xRHK points out its ***“ high practical relevance as a plug-and-play component for future UMM pipelines”***.


- **Significant Performance Gains and Great Efficiency**: Reviewer xRHK appreciates the ***“strong empirical performance”***. Reviewer ccV6 describes results on GenEval as “outstanding,” while Reviewer U3Cq also states that our 1.5B model ***“surpasses significantly larger models”*** on a few benchmarks such as GenEval and DPGBench. Reviewer uKdX and Reviewer U3Cq furthermore acknowledge the ***“realistic efficiency” of our approach which takes a mere 27 GPU-hours***.

---

### **We have conducted further experiments and made several updates to our PDF based on your valuable feedback.**

All changes are highlighted in $\textcolor{RoyalBlue}{\textbf{blue}}$ in the revised paper.


---

###  **Added more ablation studies on different data scales and data sources (Appendix F.8)**

#### **Data Scale**

| Training Data | Data Scale | GenEval  | DPG  |
| ------------- | ---------- | -------- | --------- |
| MidjourneyV6  | 10k        | 84.6     | 86.82     |
| MidjourneyV6  | 50k        | 85.3     | 87.06     |
| MidjourneyV6  | **240k**   | **85.7** | **87.21** |

**Takeaway:** RecA benefits from more data but performs strongly even with **as little as 10k images**, showing high data efficiency.

---

#### **Data Source**

| Data Source  | Quality | GenEval  | DPG  |
| ------------ | ------- | -------- | --------- |
| JourneyDB    | Medium  | 85.2     | 86.71     |
| BLIP3o-60k   | High    | 85.2     | 86.50     |
| MidjourneyV6 | High    | **85.7** | **87.21** |
| COCO2014     | Low     | 85.3     | 86.82     |

**Takeaway:** RecA is robust across **synthetic and real-world datasets**, with best performance from high-quality synthetic data.

---

###  **Added correlation between training step and overall scores (Table 5 & Appendix F.5)**

| Step     |     0    |    1k    |    2k    |    3k    |    4k    |    5k    |    6k    |    7k    |    8k    |    9k    |    10k   |
| :------- | :------: | :------: | :------: | :------: | :------: | :------: | :------: | :------: | :------: | :------: | :------: |
| **SFT**  |   72.9   |   74.7   |   73.8   |   75.4   |   74.9   |   74.8   |   74.9   |   74.0   |   73.5   |   73.7   |   73.9   |
| **RecA** | **72.9** | **82.1** | **84.1** | **85.0** | **85.0** | **85.7** | **85.7** | **85.0** | **84.7** | **84.6** | **85.0** |

**Takeaway:** RecA provides a **large immediate boost** and maintains **stable, high performance** across training steps.

---

> ### Author Response · Authors · 2025-11-22
>
> ###  **Added T2I-Compbench results in Appendix F.4**
>
> ---
>
> ### **OpenUni**
>
> | Category | Color | Shape | Texture | Spatial | Non-Spatial | Numeracy | 3D Spatial | Complex | Overall |
> | :--------------- | :------: | :-------: | :------: | :------: | :------: | :-------: | :-------: | :------: | :------: |
> | **OpenUni** |  87.47 | 75.22 | 82.87 | 78.29 | 83.10 | 64.44 | 68.60 | 82.82 | 78.84 |
> | **OpenUni-RecA** |**92.50**|**85.60**|**89.65**|**86.30**|**87.18**|**75.97**|**84.35**|**87.79**|**86.49**|
> | *vs. baseline*   | $\textcolor{ForestGreen}{+5.03}$ | $\textcolor{ForestGreen}{+10.38}$ | $\textcolor{ForestGreen}{+6.78}$ | $\textcolor{ForestGreen}{+8.01}$ | $\textcolor{ForestGreen}{+4.08}$ | $\textcolor{ForestGreen}{+11.53}$ | $\textcolor{ForestGreen}{+15.75}$ | $\textcolor{ForestGreen}{+4.97}$ | $\textcolor{ForestGreen}{+7.65}$ |
>
> **Takeaway:** RecA provides **substantial improvements** across all reasoning skills, especially Numeracy and 3D Spatial reasoning.
>
> ---
>
> ### **Harmon**
>
> | Category | Color | Shape | Texture | Spatial | Non-Spatial | Numeracy | 3D Spatial | Complex | Overall |
> | :-------------- | :------: | :------: | :------: | :------: | :------: | :-------: | :-------: | :------: | :------: |
> | **Harmon** | 89.55 | 79.09 | 86.28 | 84.98 | 82.77 | 69.19 | 75.58 | 82.05 | 81.36 |
> | **Harmon-RecA** |**93.72**|**84.32**|**91.37**|**92.37**|**87.12**|**79.78**|**86.08**|**88.51**|**88.03**|
> | *vs. baseline*  | $\textcolor{ForestGreen}{+4.17}$ | $\textcolor{ForestGreen}{+5.23}$ | $\textcolor{ForestGreen}{+5.09}$ | $\textcolor{ForestGreen}{+7.39}$ | $\textcolor{ForestGreen}{+4.35}$ | $\textcolor{ForestGreen}{+10.59}$ | $\textcolor{ForestGreen}{+10.50}$ | $\textcolor{ForestGreen}{+6.46}$ | $\textcolor{ForestGreen}{+6.67}$ |
>
> **Takeaway:** RecA significantly strengthens **all reasoning dimensions**, with the largest gains again in Numeracy and 3D Spatial.
>
> ---
>
> ### **Show-o**
>
> | Category | Color | Shape | Texture | Spatial | Non-Spatial | Numeracy | 3D Spatial | Complex | Overall |
> | :-------------- | :------: | :------: | :------: | :------: | :------: | :------: | :------: | :------: | :------: |
> | **Show-o** | 88.60 | 77.68 | 85.23 | 85.75 | **85.38** | 71.39 | 75.38 | 82.73 | 82.12 |
> | **Show-o-RecA** | **90.88** | **82.22** | **88.37** | **87.28** | 85.13 | **74.42** | **81.42** | **85.07** | **84.47** |
> | *vs. baseline* | $\textcolor{ForestGreen}{+2.28}$ | $\textcolor{ForestGreen}{+4.54}$ | $\textcolor{ForestGreen}{+3.14}$ | $\textcolor{ForestGreen}{+1.53}$ | -0.25 | $\textcolor{ForestGreen}{+3.03}$ | $\textcolor{ForestGreen}{+6.04}$ | $\textcolor{ForestGreen}{+2.34}$ | $\textcolor{ForestGreen}{+2.35}$ |
>
> **Takeaway:** RecA improves most skills, with especially strong gains in **Shape** and **3D Spatial** reasoning.
>
> ---
>
> ### **BAGEL**
>
> | Category | Color | Shape | Texture | Spatial | Non-Spatial | Numeracy | 3D Spatial | Complex | Overall |
> | :------------- | :------: | :------: | :------: | :------: | :------: | :------: | :------: | :------: | :------: |
> | **BAGEL** | 93.05 | 84.25 | 88.95 | 88.42 | 88.27 | 79.43 | 81.47 | 87.85 | 86.74 |
> | **BAGEL-RecA** |**93.62**|**85.72**|**90.58**|**88.57**|**88.80**|**81.88**|**83.50**|**88.74**|**87.89**|
> | *vs. baseline* | $\textcolor{ForestGreen}{+0.57}$ | $\textcolor{ForestGreen}{+1.47}$ | $\textcolor{ForestGreen}{+1.63}$ | $\textcolor{ForestGreen}{+0.15}$ | $\textcolor{ForestGreen}{+0.53}$ | $\textcolor{ForestGreen}{+2.45}$ | $\textcolor{ForestGreen}{+2.03}$ | $\textcolor{ForestGreen}{+0.89}$ | $\textcolor{ForestGreen}{+1.15}$ |
>
> **Takeaway:** Even on a very strong baseline, RecA consistently improves all categories.
>
>
> ---
>
>
> ---
>
> ###  **Other updates in the revised PDF**
>
> * Reclassified UMMs into **Discrete / Continuous / MAR** types.
> * Added explanation of **visual understanding embeddings** (Appendix C.1).
> * Reorganized Appendix “Discussion” → “More experimental results and analysis”.
> * Added deeper mechanism analysis in **Appendix D**.
> * Added more **reconstruction visualizations** (Appendix F.7).
> * Added **image diversity** results (Appendix J).
> * Clarified implementation details such as **input image resolution** in Sec. 2.2.
> * Extended RecA experiments to **VQGAN-based Show-o** (Appendix F.2).
>
> ---
>
> **We will open-source all code and models** to ensure full reproducibility and support future research. If there are any further questions or suggestions, we would be more than happy to address them!

---

### Author Response · Authors · 2025-12-03
**Summary of the reviewers' comments and our rebuttal.**

Dear AC,

Below, we summarize the reviewers' comments and our rebuttal for your convenience.

### Brief Summary of Our Work

We introduce **RecA**, a new self-supervised post-training approach to enhance the generation and editing ability of Unified Multimodal Models (UMMs) without any text captions or external models. RecA received a high initial average rating of 6.5 **(8, 8, 6, 4)**.

### Key Strengths Highlighted by Reviewers

RecA is recognized by all reviewers:

-  **Intuitive and Effective**: "intuitive and well-motivated" (ccV6), "strong empirical performance" (xRHK), "surprisingly works well" (U3Cq), and improves performance "without consuming any extra labeled or distilled data" (uKdX).
- **Promising Generality and Applicability**: "architecture-agnostic" (U3Cq), "widely applicable" (ccV6), "high practical relevance as a plug-and-play component for future UMM pipelines" (xRHK), and "consistently improves the fidelity" (uKdX).
- **Significant Performance Gains and Great Efficiency**: "outstanding" results on GenEval (ccV6), our 1.5B model "surpasses significantly larger models" on benchmarks like GenEval and DPGBench (U3Cq), with "realistic and practical efficiency" taking only 27 GPU-hours (uKdX, U3Cq).

### Detailed Rebuttal Responses

We sincerely appreciate the valuable feedback from all reviewers, and we believe the additional experiments conducted during the rebuttal further demonstrate the robustness and efficiency of RecA. Mainly:

- For **Reviewer ccV6**, we verified RecA's robustness across different datasets and demonstrated its scaling potential through data scaling ablations.
- For **Reviewer xRHK**, we evaluated our model on the more complex and diverse **T2I-CompBench**, achieving significant improvements that prove our method is effective, generalizable, and versatile.

Below, we primarily wish to highlight our responses to the concerns raised by **Reviewer uKdX** and **Reviewer U3Cq**, as we believe these have been comprehensively resolved.

**Reviewer uKdX**

- The reviewer questioned how RecA could improve UMM’s understanding capability and criticized that we have no direct evaluation on understanding benchmarks. We clarified this misunderstanding: RecA is **designed to improve UMM’s generation and editing capabilities**. And in our paper, we have already shown that **understanding capabilities are preserved and not degraded**.
- The reviewer noted a lack of deep analysis. In response, we added a comprehensive mechanistic analysis on (Appendix D). This section deeply analyzes the core principles behind RecA, providing the "final piece of the puzzle" for our methodology.
- The reviewer requested details on “how visual understanding embeddings are integrated”. We have added specific descriptions in **Section 2.2** and detailed illustrations in **Appendix C** to ensure full reproducibility.
- The reviewer is concerned about "Hyperparameter Sensitivity". We demonstrated that RecA is robust to hyperparameters. Except for BAGEL (adjusted due to limited global batch size from compute constraints), all models use **default hyperparameters** from their official repositories. We added: (I) ablations using default LR for BAGEL, and (II) training curves (Table 5, Figure 16) to prove stability.
- The reviewer viewed "post-training" as a limitation. We respectfully argue it is a **strength**: RecA saves massive compute by avoiding retraining and is plug-and-play for existing UMMs. Regarding data scalability, we added **data scaling ablations (Appendix F.8)** showing clear potential for improvement with more data. While pre-training is a promising future direction, it is beyond the scope of this efficient post-training study.

**Reviewer U3Cq**

- The reviewer expressed concern about potential artifacts or degradation in diversity. In response, we added **uncurated generated samples (Appendix J)**. These qualitative results demonstrate that RecA maintains high visual quality and diversity.
- Similar to uKdX, we provided the deep mechanistic analysis in **Appendix D**.
- The reviewer asked whether RecA can be combined with RL. We conducted extensive experiments applying RecA to RL-tuned UMMs (Mask-GRPO, HermesFlow). Results show that RecA can be combined with RL for **complementary enhancement**. We also showed that CoT reasoning capabilities in BAGEL are improved zero-shot after RecA training.

Regrettably, the premature conclusion of the discussion prevented further feedback and re-evaluation from other reviewers. However, we are encouraged that Reviewer ccV6 has explicitly confirmed our rebuttal's effectiveness and **"insists on a clear acceptance"**.

Finally, we also commit to **open-sourcing our code and post-trained model weights** so the community can verify and follow our results. For further details, please refer to our **global rebuttal** and **individual responses** to each reviewer. We sincerely appreciate your time and effort in handling our manuscript.

Authors

---

### Meta-Review · Area_Chair_fyUw · 2026-01-06

**Summary:**

This paper presents Reconstruction Alignment (RecA), a simple but effective post-training technique for unified multimodal models (UMMs). The paper received an overall positive rating (8, 8, 6, 4). Reviewer uKdX gave a rating of 4, mainly raising concerns about the evaluation on understanding benchmarks, a lack of deep analysis, and missing experiments on hyperparameters. However, the other reviewers (ccV6, xRHK, U3Cq) recognized the method's strong performance and efficiency.The authors provided a thorough rebuttal, which well addressed the questions raised by Reviewer uKdX.

After carefully reading the paper, reviewer comments, and the author feedback, the AC agree with Reviewers ccV6, xRHK, and U3Cq that this paper can be accepted for ICLR 2026. I believe the concerns raised by Reviewer uKdX have been addressed satisfactorily. The authors should include all the discussions and the additional experimental results in the camera-ready version.

**Reviewer Concerns:**

Their major concerns regarding the model robustness across datasets, scaling potential, and evaluation on real-world prompts raised by reviewer ccV6 and xRHK were comprehensively addressed.

The request from reviewer U3Cq for a stronger theoretical grounding and the analysis of failure cases was addressed during rebuttal.

The reviewer uKdX's concerns regarding the lack of evaluation on understanding benchmarks, deep analysis, and missing experimental details have been well addressed.

Based on the rebuttal and the additional experiments provided, I believe there are no major outstanding concerns.

**Reviewer Scores:**

Given that the authors have addressed all raised concerns, it is likely that Reviewers ccV6, xRHK, and U3Cq would maintain their ratings, while Reviewer uKdX would increase the score to a positive rating.

---

### Decision · Program_Chairs · 2026-01-26

Accept (Poster)